# Alternative molecular mechanisms for force transmission at adherens junctions via β-catenin-vinculin interaction

Nicole Morales-Camilo[1], Jingzhun Liu [2], Manuel J. Ramírez[1,3], Patricio Canales-Salgado [4,5], Juan José Alegría [4,6], Xuyao Liu[2,7], Hui Ting Ong[7], Nelson P. Barrera [3], Angélica Fierro[8], Yusuke Toyama [7,9], Benjamin T. Goult [10,11], Yilin Wang[7], Yue Meng[7], Ryosuke Nishimura [7], Kedsarin Fong-Ngern[7], Christine Siok Lan Low[7], Pakorn Kanchanawong [7,12], Jie Yan [2,7], Andrea Ravasio [4] ✉ & Cristina Bertocchi [1,13] ✉

Force transmission through adherens junctions (AJs) is crucial for multicellular organization, wound healing and tissue regeneration. Recent studies shed light on the molecular mechanisms of mechanotransduction at the AJs. However, the canonical model fails to explain force transmission when essential proteins of the mechanotransduction module are mutated or missing. Here, we demonstrate that, in absence of α-catenin, β-catenin can directly and functionally interact with vinculin in its open conformation, bearing physiological forces. Furthermore, we found that β-catenin can prevent vinculin auto-inhibition in the presence of α-catenin by occupying vinculin´s head-tail interaction site, thus preserving force transmission capability. Taken together, our findings suggest a multi-step force transmission process at AJs, where α-catenin and β-catenin can alternatively and cooperatively interact with vinculin. This can explain the graded responses needed to maintain tissue mechanical homeostasis and, importantly, unveils a force-bearing mechanism involving β-catenin and extended vinculin that can potentially explain the underlying process enabling collective invasion of metastatic cells lacking α-catenin.

Multicellularity is an evolutionary milestone that enabled the richness of life seen in nature. While plant cells are immobilized within a thick matrix of cellulose, animal cells maintain a remarkable motility even if they aggregate to form multicellular organisms. Such cellular motility is necessary to fulfill complex functions such as embryonic development, immunosurveillance and tissue regeneration, and it contributes to pathological states such as cancer metastasis[1]. For this reason, cells require adhesive mechanisms that are strong enough to preserve tissue integrity, and sufficiently dynamic to allow for cell movement. Hence, it is important to understand how adhesive complexes connecting cells dynamically and robustly solve this apparent mechanical paradox.

Within epithelial tissues, forces between cells are borne by specialized structures at cell-cell contacts, named Adherens Junctions (AJs)[2]. The propagation of such forces is essential in morphogenesis, tissue homeostasis, and collective cell migration[3–6], and thus it is cell type specific, and it is subject to context-specific spatiotemporal regulations[7]. AJs are prominent adhesive structures in epithelia that physically couple neighboring cells via Ca[2+]-dependent homophilic interaction between the extracellular domains of classical cadherins[8–10]. Cadherin receptors are, in turn, coupled to the acto-myosin cytoskeleton via an extensive ensemble of cytoplasmic proteins[11,12], thereby integrating the mechanobiological processes of

cell-cell adhesions with a variety of important cellular pathways[13]. In recent years, we gained a better understanding of the functions of AJs across multiple length scales, spanning from molecular to tissue levels[4]. However, structural aspects of the nanoscale organization and molecular mechanics of AJ proteins into modular complexes have been only recently explored[14,15]. Hence, processes occurring at this length scale, including how mechanical force transmissions are mediated and regulated through AJs, is unclear.

The minimal cadherin/catenin complex is widely considered the main conduit of mechanical force between epithelial cells[16]. The coupling of actomyosin contractility and intramolecular tension in E-cadherin and α-catenin has been demonstrated using Forster Resonance Energy Transfer (FRET) probes[17–19] and optical trap-based assays[16,20]. Moreover, various studies including single-molecule force spectroscopy revealed that tension-dependent conformational changes of α-catenin control the recruitment of vinculin within the complex, thus reinforcing the connection between AJs and the actin cytoskeleton[21–24]. Furthermore, our group demonstrated by super-resolution microscopy that AJ complexes are multilayered with a cadherin/catenin compartment proximal to the plasma membrane and vinculin being centrally located at the interface between the cadherin/catenin compartment and the actin cytoskeleton[14]. The conformational landscape of vinculin is complex. Previous studies based on vinculin tension and conformational FRET sensors[14,25,26] have shown that, while cytosolic vinculin is in a closed auto-inhibited conformation, vinculin can be primed at the adhesions by talin or α-catenin. This event induces a partial opening of vinculin into a compact conformation at low-tensional states. Upon mechanical and biochemical signals, vinculin was found to undergo conformational extension, which, in turn, modulates the AJ nanostructure as well as regulates transmission of mechanical tension, consistent with the molecular clutch model[25]. Hence, current consensus is that one mode of action passing through the interaction between α-catenin and vinculin is responsible for force transmission. However, this does not well reconcile with the multi-functional and context-dependent roles of the many AJ components[26,27], the wealth of intra-AJ interaction partners[6] and the intrinsic high-density organization and dynamic plasticity of AJ. Similarly, understanding of how the components of the AJ function as an integrated system remain elusive. Recent studies demonstrated a hierarchical organization of the force transmission module through a dominant tether formed by E-cadherin/β-catenin/α-catenin/vinculin/actin[16,28]. However, several alternative proteins may also participate in connecting cadherin with actin, including myosin VI[29], eplin[30], α-actinin[31], and afadin[32,33]. Additionally, β-catenin has been proposed to directly interact with vinculin[34,35], which could potentially form an alternative bypass connection between cadherin and the actin cytoskeleton. Collectively, these observations suggest that in AJs a mixture of different mechanical connectors, rather than a single dominant tether, may couple the cadherin receptors and actomyosin cytoskeleton. Furthermore, due to the vital importance of AJs, alternative structural redundancies would fulfill the theoretical requirements expected for a robust system[36]. Importantly, alternative mechanisms for force transmission at the AJs would also explain how various cancer cell types achieve collective migration and invasion even in absence of α-catenin[37].

In this study, we used molecular modeling, high- and super-resolution microscopy associated with engineered biomimetic substrates, laser nanoscissors, live-cell imaging and biochemistry tools to investigate possible alternative conformational arrangement of the catenins/vinculin complex mediating force transmission at the AJ. Importantly, we found that, in its active conformation, vinculin can form a mechanical connection with β-catenin, possibly bypassing α-catenin. However, α-catenin may be required for initial vinculin activation, as suggested by the weak interaction between vinculin in its compact conformation and α-catenin. Taken together our result

demonstrate that force transmission at the AJs is a multi-step mechanochemical process able to provide graded engagement between cells and to maintain connection in absence of α-catenin, as observed in some collectively invading cancer[37,38].

## Results

### Vinculin forms a stable interaction with α-catenin VBS at S1 and S2 interfaces

The canonical model of mechanotransduction at AJ postulates that α-catenin constitutively recruits vinculin to strengthen force-response therein and maintain cell cohesiveness within epithelial tissues[39]. The molecular interfaces involved in the interaction and stabilization between vinculin and α-catenin, we performed molecular modeling using the three-dimensional descriptions for vinculin and α-catenin obtained by AlphaFold[40,41] database and the crystal structures information from Protein Data Bank (PDB).

A full-length vinculin structure from AlphaFold database indicating the four domains (called D1-D4) in an auto-inhibited conformation with the tail (Vt) is represented in Fig. 1a. The D1 domain (residues 1-252 highlighted in cyan) possesses three potential interaction sites called S1, S2 and S3 (Fig. 1b). S1 is located in the groove formed between α-helix 1 and 2 (corresponding to VD1a in ref. 42) and it has been reported to take part in the 5-helix bundle binding mechanism for the interaction with proteins such as talin, α-actinin, IpaA and α-catenin. S2 (corresponding to VD1b in ref. 42) is positioned between α-helix 4 and 5 and it has been suggested to be involved in a transient interaction with α-catenin. Finally, S3 is between α-helix 1 and 4 and it promotes in the head-tail intramolecular interaction and pivot for vinculin unfolding mechanism[43].

The interaction between vinculin and α-catenin has been previously characterized[23,24,42,44]. In Fig. 1c the full-length α-catenin structure from AlphaFold database is shown, with its VBS (vinculin binding site) segment (residues 305–355) highlighted in green. The extended α-catenin VBS structure from crystal structure (PDB code 4EHP)[23] at 2.6 Å of resolution is represented in Fig. 1d. As shown in the alignment with the amino acid sequences of other known VBS ligands (i.e., VBS1, VBS2 and VBS3 of talin and VBS of α-actinin, Supplementary Fig. 1), VBS from α-catenin presents a similar conserved hydrophobic motif of aliphatic residues (L348, L347, L344, V340, and I333) that is suitable for interaction with S1 (LLLVI in Fig. 1d). To represent the molecular interfaces involved in the interaction between vinculin and α-catenin, we simplified the system and focused our structural studies on the vinculin D1 domain (VD1) and α-catenin VBS short peptides. For simplicity, from here onward we will refer to vinculin D1, simply with vinculin or with the name of the mutation (vinculin T12, vinculin head, vinculin A50I).

During 1 µs of molecular dynamics (MD) simulation, α-catenin VBS maintained an interaction with S1 and S2 of vinculin (Fig. 1e, Supplementary Movie 1 and Supplementary Fig. 2a). This agrees with previous single molecule magnetic tweezer studies[44] from authors of this paper that experimentally demonstrated the viability of the formation of a complex between α-catenin VBS and S1 (VD1a) and S2 (VD1b) of vinculin. While the experimental verification of the exact residues mediating this interaction is still missing, MD analysis for the stability of the complex predicts the involvement of intermolecular hydrogen bonds (S349, Y351, S323, E336 of α-catenin and Q19, R105, K170, N53 of vinculin), hydrophobic stabilization of conserved non-polar residues motif inside the groove of S1, and salt bridge interactions (E60, E66 of vinculin and R332, R326 of α-catenin) (Fig. 1e). Furthermore, Adaptive Poisson-Boltzmann Solver (APBS) analysis of the electrostatic surface potential shows a high negative charged area (red) on vinculin between S1 and S2 and, a low electron density area (blue) on the central part of α-catenin VBS (Fig. 1f). Based on the general interactions in the dimer vinculin/α-catenin, we performed a per-residue Molecular Mechanics with Poisson−Boltzmann and Surface Area solvation (MM-PBSA) analysis to calculate the global

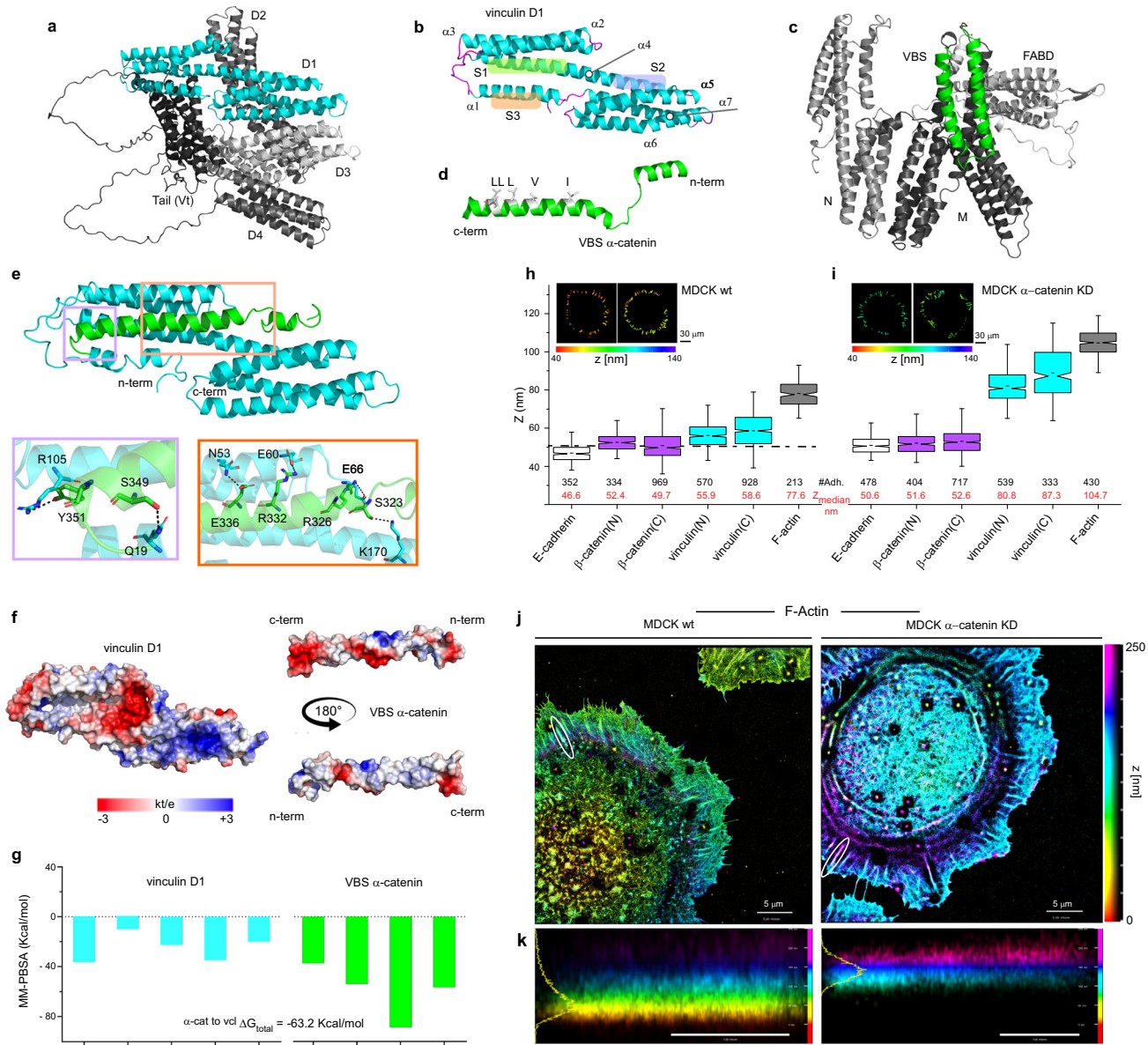

**Fig. 1 | Compact vinculin loses its contact with the cadherin-catenin layer when α-catenin is depleted. a** Cartoon representation of full-length vinculin structure from AlphaFold database indicating the four domains (called D1-D4) in an auto-inhibited conformation with the tail (Vt). **b** Cartoon representation of vinculin D1 domain showing the three important binding sites (herein referred to as S1, S2, and S3) for vinculin function. **c** Cartooon representation of full-length α-catenin structure obtained from AlphaFold database. In different gray scales the N, M, and FABD domains are indicated, highlighting in green the vinculin binding site (VBS) segment (amino acids 305−355). **d** Cartoon representation of α-catenin VBS showing polar non-charged amino acid in green and the non-polar residues L348, L347, L344, V340, and I333 in white. These residues are those that partially match the conserved hydrophobic motif of vinculin binding site (VBS) segments LxxAAxxVAxxVxxLIxxA (alignment in Supplementary Fig. 1) previously reported by Izard T. and coworkers for VBS1, VBS2, and VBS3[89]. **e** Cartoon representation of the MD simulation last frame of vinculin D1 (cyan) and αcatenin VBS (green) complex. The two insets illustrate the interfaces of the main surface interactions: inter-molecular hydrogen bonds (S349, Y351, S323, E336 of α-catenin and Q19, R105, K170, N53 of vinculin), hydrophobic stabilization of conserved non-polar residues motif inside the groove of S1, and salt bridge interactions (E60, E66 of vinculin and R332, R326 of α-catenin). **f** Electrostatic surface potential of the vinculin D1 and α-catenin VBS contact interfaces. **g** MM-PBSA per-residue energetic contribution. The

energetic contribution of arginine residues (R326 and R332) of α-catenin together with asparagine (N53) and glutamic acid (E60) residues of vinculin coupled with the short mean distance of interaction in N53/R326 and E60/R332 of close to 3 Å along the MD (Supplementary Fig. 3) support the involvement of these residues in the complex stabilization. Notched box and whisker plots indicating the median $z$centre position of indicated proteins in control MDCK (**h**) and α-catenin KD MDCK (**i**) cells obtained by SAIM (statical analysis reported in Supplementary Table 1). Box represents median, 1st, and 3rd quartiles; whiskers, 5th and 95th percentiles; #Adh. indicates the number of adhesions measured. Dotted line in (**h**) corresponds to the C-terminal z-position of α-catenin ($z$centre = 51.95 nm). Inserts in (**h**) and (**i**) indicate Topographic Map of the z-position of vinculin n- and c- termini. MDCK wt and MDCK α-catenin KD expressing FP probes of vinculin n- and c-terminus (left and right, respectively) were cultured on biomimetic E-cadherin-Fc -coated (Supplementary Fig. 4a) silicon wafer and imaged by surface-generated structured illumination microscopy. Color bar, 40−140 nm. Scale Bar, 10 μm. **j, k** iPALM imaging for F-actin in MDCK wt (left) and when α-catenin is depleted in MDCK α-catenin KD (right) in cells on E-cadherin biomimetic substrate. F-actin in MDCK cell labeled by Alexa Fluor 647-phalloidin. **j** top view, and (**k**) side view of white box in (**j**). Colors (hue scale in I, 0−150 nm) indicate the vertical (z) coordinate, relative to the sub-strate surface (z = 0 nm, red). Scale bars: 1 μm (**j**), 250 nm (**k**). Associated plot in Supplementary Fig. 4h.

free energy ($\Delta G_{vinculin/\alpha\text{-}catenin}$ = −63.2 Kcal/mol) of the dimer (the residues that have the largest energetic contribution to the complex stabilization are represented in Fig. 1g and the short mean distance along the MD in Supplementary Fig. 3). These results are in line with previously published papers by W. Weis[42] and our groups[44], and support the involvement of S1 (VD1a) and S2 (VD1b) in the interaction with α-catenin VBS.

## α-catenin depletion induces vinculin disengagement from the cadherin/catenin layer

In previous studies, we demonstrated the use of a model system where cells seeded on a cadherin biomimetic substrate allow for better optical accessibility for high-resolution imaging of cadherin-based adhesions[14,45] (Supplementary Fig. 4a). The use of such a platform, in conjunction with surface generated structured illumination microscopy Scanning Angle Interference Microscopy (SAIM)[46–48], permitted sub-20 nm precision analysis of position and orientation of proteins along the vertical (z)-axis perpendicular to the plasma membrane. This led to the identification of the nanoscale architecture and the molecular mechanics therein[14].

The same strategy was here employed to investigate the dependence of the nanoscale architecture of E-cadherin-based adhesions on α-catenin. As shown in Fig. 1h, i, we compared the nanoscale organization of the vinculin/catenins complex in control epithelial cells (MDCK wt) and α-catenin knockdown cell line[49] (MDCK α-catenin KD). Expression levels of α-catenin, β-catenin and vinculin in these cell lines are shown in Supplementary Fig. 4b−e and immunofluorescence in Supplementary Fig. 4f−g demonstrated a high degree of colocalization of endogenous vinculin and β-catenin in MDCK wt as well as MDCK α-catenin KD. The fluorophore z-position relative to the substrate surface ($z = 0$ nm) was analyzed pixel-wise, with the median value, $^z$center, for adhesion regions of interest (ROIs) used as the representative protein z-position. We found that E-cadherin (C-terminal GFP fusion) was localized at similar z-positions in MDCK wt and α-catenin KD ($^z$center = 46.56 nm and 50.57 nm, respectively). In contrast, a drastic difference was observed in the z-position of vinculin. As shown in Fig. 1h, i (Supplementary Tables 1, 2) in MDCK wt vinculin was positioned proximal to the cadherin/catenin compartment (vinculin N-terminus at $^z$center = 55.93 nm, compared to 58.61 nm for the C-terminus), where α-catenin assumes an open conformation, as previously demonstrated by α18 immunostaining and FRET measurements with α-catenin conformational sensors[14]. However, upon α-catenin knock down (90% ca. of α-catenin depletion), vinculin remained associated with the adhesion, but moved 25–30 nm upwards (vinculin N-terminus $^z$center = 80.82 nm and C-terminus = 87.30 nm) (Fig. 1i), probably in association with actin or actin-regulatory proteins such as VASP or α-actinin[50,51]. Using iPALM superresolution microscopy, we confirmed that, when α-catenin is depleted, junctional F-actin localizes at a z-position similar to that observed for vinculin upon α-catenin depletion ($^z$center > 80nm−Fig. 1j, k and Supplementary Fig. 4h). This confirmed previous observations[14] that in MDCK wt α-catenin is in an open conformation, stretched to maintain the connection between the cadherin/catenin layer and F-actin, and with an exposed VBS for vinculin engagement. At reduced α-catenin expression, F-actin is no longer maintained in position by the connection to α-catenin and shifts upwards ($^z$center is 25 nm higher) together with vinculin, further away from the cadherin-catenin layer. Importantly, we have previously observed that upon re-expression of α-catenin vinculin restores its central position at the interface zone which suggests that α-catenin may play the role of anchor for attachment of vinculin with the cadherins[14].

## α-catenin depletion and relief of autoinhibition of vinculin allow for vinculin/β-catenin interaction

Since vinculin can switch between auto-inhibited (closed), compact and extended conformations[52], the small difference in N- and C-termini z-position observed in MDCK α-catenin KD (Fig. 1i) suggests that vinculin is in a closed conformation. To confirm this hypothesis, we used a constitutively active vinculin mutant (vinculin-T12), which contains four mutated residues in the vinculin tail domain (D974A, K975A, R976A, and R978A) that reduce the intermolecular head to tail affinity[53]. Interestingly, when we expressed vinculin T12 in MDCK α-catenin KD cells, we observed a highly polarized orientation of vinculin effectively spanning between the cadherin/catenin and actin compartments (vinculin-T12 N-terminus at $^z$centre = 51.67 nm, and 82.30 nm for the C-terminus−Fig. 2a). This indicates that, despite the significant depletion of α-catenin (Supplementary Fig. 4b, c), vinculin in this conformation could still localize at cadherin-mediated adhesions (Fig. 2b and Supplementary Fig. 5) to form a physical connection between the cadherin- catenin compartment and the actin cytoskeleton. This was verified in both α-catenin knockdown as well as knockout conditions confirming that the recruitment of active vinculin to the cadherin-catenin layer is not due to the minimal amount of leftover α-catenin present in the MDCK α-catenin KD (Supplementary Figs. 5, 6). We corroborated this observation by assessing the localization of the vinculin head domain alone at the adhesion (Supplementary Fig. 4j) and by probing for its z-position (Supplementary Fig. 4i). Vinculin head z-position was found to be lower, proximal to the cadherin/catenin compartment (vinculin head N-terminus at $^z$centre = 57.29 nm, and 68.45 nm for the C-terminus), similar to that of vinculin wt in MDCK wt ($^z$centre = 55.93 nm and 58.61 nm for N- and C-terminus, respectively). These results suggest that once relieved of the autoinhibitory head-tail interaction, the vinculin head domain of vinculin is capable of interacting with a partner within the cadherin/catenin compartment other than α-catenin. An obvious potential candidate could be β-catenin. We observed the z-position of β-catenin ($^z$centre = 51.63 nm and 52.64 nm for N- and C-terminus, respectively, as in Fig. 1h, i, Supplementary Table 1) appears to coincide with the N-terminal z-position of vinculin-T12 or vinculin-head (Fig. 2a and Supplementary Fig. 4i) in MDCK α-catenin KD. These observations raised an interesting possibility of a direct interaction between β-catenin and vinculin in its opened conformation. We verified the interaction between VBS β-catenin/vinculin in vitro by performing co-immunoprecipitation of vinculin wt or mutants (GFP conjugated) with β-catenin from MDCK wt and when α-catenin is depleted in MDCK α-catenin KD (α-catenin expression levels in MDCK α-catenin KD are provided in Supplementary Fig. 4b−d). In MDCK wt, significant co-precipitation was observed between vinculin wt and β-catenin (Fig. 2c, d). This was significantly diminished to baseline level in MDCK α-catenin KD. This is in line with our SAIM results (Fig. 2a) that show a detachment of vinculin from the cadherin/catenin layer when α-catenin is depleted. In contrast, when vinculin-T12 was used for the pull-down, an increase in β-catenin co-precipitation was observed, in agreement with a highly polarized orientation for constitutively active vinculin (vinculin-T12) that effectively spans between the cadherin/catenin and actin compartments. Altogether, these data indicate that when α-catenin is low, the interaction between vinculin wt and β-catenin is minimized (effect not related to a decreased expression level of β-catenin, as indicate in Supplementary Fig. 4b and shown in the immunostaining in Fig. 2b). Nevertheless, the physical connection between cadherin/catenin layer and actin can be re-established by activated (opened) vinculin, probably through its connection with β-catenin.

## β-catenin and α-catenin can bind to the same groove on vinculin

Interaction between β-catenin and vinculin has been previously suggested to stabilize E-cadherin at the cell surface[34]. However, to date, there is no information about specific vinculin conformations (e.g., auto-inhibited, compact, unfurled, etc.) capable of interacting with β-catenin. This is probably due to the limited information about the three-dimensional structure of the N-terminal region of β-catenin that was previously suggested to contain binding site for vinculin, homologous with the amphipathic helices of talin, α-actinin, and α-catenin[34].

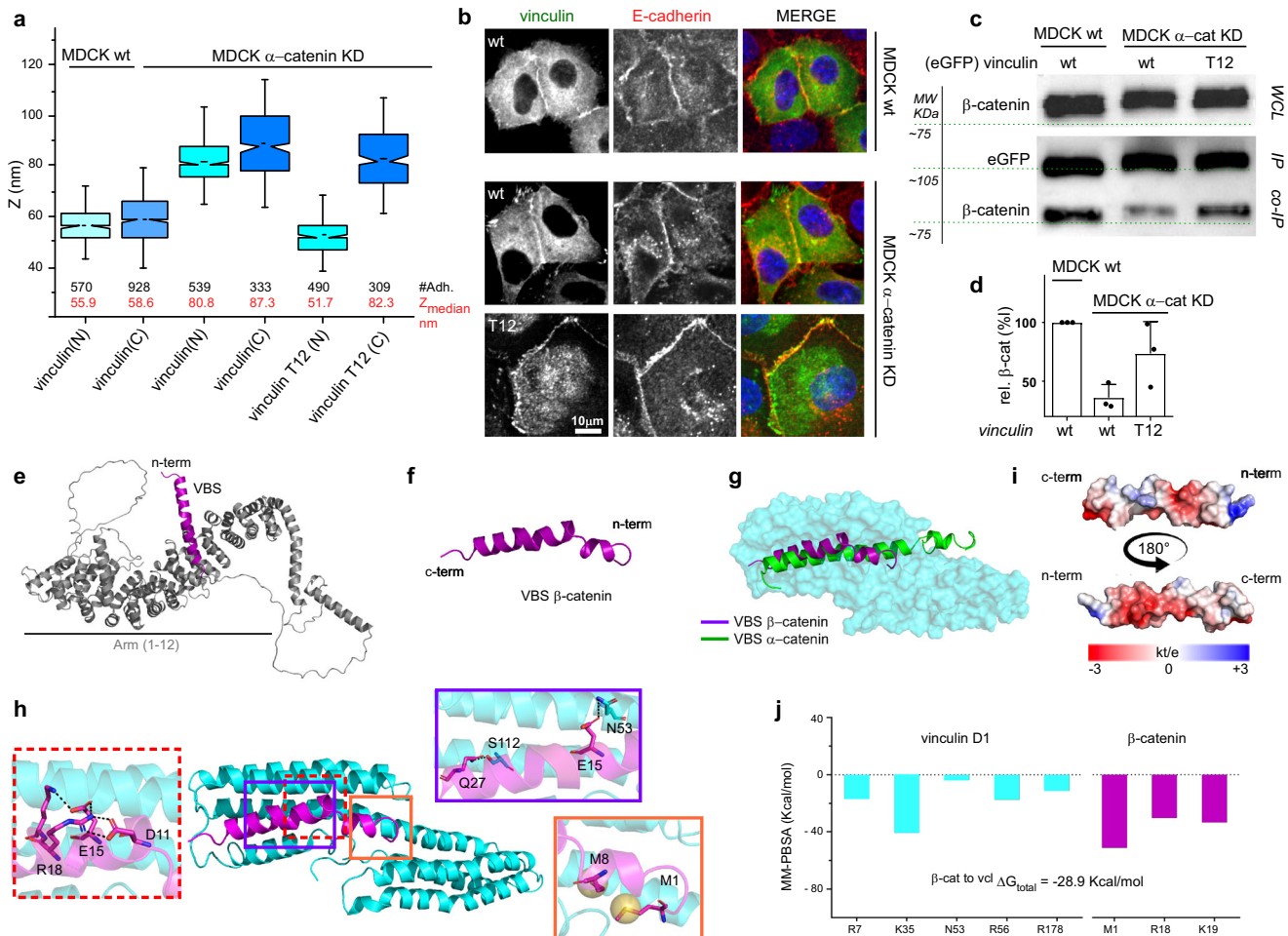

**Fig. 2 | Activated vinculin maintains its connection to the cadherin-catenin layer when α-catenin is depleted.** Activated vinculin spans the core of cadherin-based adhesions. **a** Notched box and whisker plots indicating the median zcentre position of vinculin wt and constitutively active vinculin T12 mutants in MDCK wt and α-catenin KD, obtained by SAIM (statical analysis reported in Supplementary Table 1). Box represents median, 1st, and 3rd quartiles; whiskers, 5th and 95th percentiles; #Adh. indicates the number of adhesions measured. **b** Localization of vinculin constructs to AJs in MDCK wt and MDCK α-catenin KD in monolayer. Maximum intensity projections of vinculin (GFP, green channel) and E-cadherin (antibody probes, red channel), and merged image. Scale bar, 10 μm. **c, d** Co-immunoprecipitation of β-catenin with vinculin. Representative western blots (**c**), probing for β-catenin pulled-down with GFP-vinculin constructs as indicated on the top lane and (**d**) quantification of β-catenin intensity and GFP (correspondent to

vinculin) from 3 independent experiments ($n = 3$), expressed as mean ± SD. Control for whole cell lysate (WCL) included. **e, f** Cartooon representation of full-length β-catenin structure obtained from AlphaFold database. VBS segment (amino acids 1–30) is highlighted in purple. **g** Cartoon representation of α-catenin VBS (green) and β-catenin VBS (purple) which both overlap on S1 binding site of vinculin D1 domain (cyan). **h** Cartoon representation of the MD simulation last frame of vinculin D1 (cyan) and βcatenin VBS (purple) complex. Insets with solid lines represent the residue interface of the main surface interaction and the inset with dotted line show the β-catenin intermolecular interactions. **i** Electrostatic surface potential (right) of the β-catenin VBS (cartoon representation shown on the left) contact interface with vinculin (not shown). **j** MM-PBSA per-residue energetic contribution for main residues of vinculin and βcatenin pair residue.

Thus, we set up to explore this possibility in depth. From AlphaFold database, we obtained the model indicated in Fig. 2e, where the physicochemical properties of the N-terminal segment, (residues 1–30 highlighted in purple) describe an amphipathic helical structure (Fig. 2f). The comparison of this sequence with the amino acid sequence of the VBS from α-catenin reveals 20% identity due to the presence of conserved amino acids (Q4, D6, L7, M8, M12, A13, M14, E15, P16, K19, A21, V22, and S23−Supplementary Fig. 7). These results indicate that the N-terminal of β-catenin is homologous to the amphipathic helices of known interacting partners of vinculin such as α-catenin, talin, α-actinin, IpaA[34,42]. Thus, using α-catenin VBS from the crystal structure PDB code 4EHP[23] as template (Fig. 2g) we built a three-dimensional description of vinculin/β-catenin dimer inserting the α-helix of β-catenin VBS in the S1. This complex has been analyzed by MD simulations for 1 μs (last frame in Fig. 2h and Supplementary Movie 2) and structural distances measured by Root Mean Square Deviation

(RMSD) (Supplementary Fig. 2b). The stability of vinculin/β-catenin complex during the simulation was mainly driven by interactions between N53/E15 and S112/Q27 of vinculin and β-catenin VBS, respectively (insets in Fig. 2h and Supplementary Fig. 8). Distances between these couples (3.8 Å and 3.1 Å, respectively) are larger than those corresponding to strong interactions (<2.5 Å). This could be explained by the intramolecular rearrangement of β-catenin VBS at the beginning of simulation (50 ns), resulted from Coulombic interactions (D11, E15, R18, K19) that induce a displacement of M1 and M8 of β-catenin VBS from the S1 hydrophobic pocket of vinculin (inset in Fig. 2h). This displacement is also associated with distancing between E130 of vinculin and D6 and E9 of β-catenin, again resulting from Coulombic repulsions (Supplementary Fig. 9). Such electrostatic repulsions contribute to the observed energy disparities in the binding process, resulting in higher binding energies as compared to the vinculin/α-catenin dimer. In Fig. 2i, we report the final electrostatic distribution

map (APBS) of β-catenin VBS after MD simulation showing the radical change in surface charges compared with those found in α-catenin VBS (Fig. 1f). A MM-PBSA analysis resulted in a favorable free energy ($\Delta G_{vinculin/\beta\text{-catenin}}$ = −28.9 Kcal/mol) for the complex where the best contribution per residue were R7, K35, N53, R56, R178 in vinculin and M1, R18, and K19 in β-catenin (Fig. 2j). In conclusion, our results indicate an energetically favorable interaction of β-catenin VBS with vinculin S1 through the canonical 5-helix bundle mechanism[42].

## A50I mutation in vinculin destabilizes VBS β-catenin/vinculin interaction

Similarly to the effect on the interaction with talin, α-actinin and IpaA[52–54], a mutation of alanine-to-isoleucine mutated (A50I) in vinculin has been previously suggested to affect the binding to β-catenin[34]. Co-immunoprecipitation data (Supplementary Fig. 10), SAIM results (Fig. 3a), and analysis of protein recruitment at the adhesion (Supplementary Fig. 5b) confirmed loss of interaction between vinculin T12 and β-catenin in the presence of A50I mutation (vinculin T12-A50I) in MDCK α-catenin KD cells. This was also supported by the weak localization of this mutant at cell-cell adhesion (Fig. 3b). To describe the structural implications on this vinculin mutation in the protein-protein interaction, we carried out 1 μs MD simulations for the dimer formed between vinculin A50I and β-catenin VBS (Supplementary Movie 3). Our results showed a displacement of β-catenin VBS from vinculin S1 (Fig. 3c) and shortening of the distance between α-helix 1 and α-helix 2 of vinculin A50I from ~12 Å to ~6 Å at 800 ns, thus effectively closing the grove (Supplementary Fig. 11) and impeding β-catenin VBS positioning into the 4-helices pocket of vinculin. Furthermore, it is possible to observe the generation of a network of intramolecular interactions between the residues D11, E15, R18, and K19 of β-catenin (inset of Fig. 3c). In accordance with the hypothesized destabilization of the complex produced by the mutation, total energy obtained using MM-PBSA indicated a statistically significant ($p = 0.0028$) lower affinity for vinculin A50I/β-catenin VBS as compared to vinculin wt/β-catenin VBS ($\Delta G_{vinculin/\beta\text{-catenin}}$ = −28.9 Kcal/mol vs $\Delta G_{vinculin(A50I)/\beta\text{-catenin}}$ = −25.5 Kcal/mol—Fig. 2i, Fig. 3d and Supplementary Tables 8, 9).

Interestingly, some authors report that under equilibrium solution conditions, the isoform E (epithelial) of α-catenin does not interact with the A50I mutant[42], whereas others indicate that the presence of A50I mutation leaves the interaction of vinculin with α-catenin unperturbed[34]. In an effort to address this, we performed MD for vinculin A50I/ α-catenin (Supplementary Movie 4). As shown in Fig. 3e, salt bridges (E60/R332 and E181/R330) and hydrogen bonds (N53/E336 and Q19/S349) mediate the interaction between vinculin A50I and α-catenin (insets in Fig. 3e). As mentioned earlier, it has been proposed that α-catenin can be potentially involved in a transient interaction with S2 in addition to S1 (Figs. 1b, 3E yellow inset)[42]. Our calculations show a strong effect of the A50I mutation on the binding affinity of α-catenin with S2 in comparison to S1. An intramolecular interaction between R326 and D321 residues of α-catenin led to a conformational rearrangement that induced the unfolding of α-helix conformation of that segment. However, the interaction was maintained through M319 residue in S2. When looking at the free energy, MM-PBSA showed that the affinity between vinculin/α-catenin is maintained in the presence of the A50I mutation ($\Delta G_{vinculin/\alpha\text{-catenin}}$ = −63.2 Kcal/mol vs $\Delta G_{vinculin(A50I)/\alpha\text{-catenin}}$ = −66.0 Kcal/mol; $p = 0.0927$—Supplementary Tables 8, 9) supporting the hypothesis that α-catenin interacting with vinculin is not globally hindered by the A50I mutation (Fig. 3f). This site of possible interaction between vinculin and α-catenin involving S2 binding site could help reconcile divergent previous findings[33,39].

To corroborate the structural results for vinculin A50I/catenin complexes we compared the effect of two known mutations for α-catenin (L344P) and β-catenin (M8P), respectively, which have been previously reported to influence the interaction with vinculin[34,55]. During MD, both mutations displayed drastic changes in vinculin/

catenin binding modes (Supplementary Movies 5, 6). Vinculin/β-catenin(M8P) showed a behavior astonishingly similar to that described for vinculin (A50I)/β-catenin (Fig. 3g). After 700 ns, the groove where lies S1 in vinculin closed, approaching a distance of 8 Å between α-helices 1 and 2 and, consequently, β-catenin(M8P) moved away from the groove (Supplementary Fig. 12). Not surprisingly, the free energy calculated by MM-PBSA and the per-residue contribution (Fig. 3h) are very similar with those obtained for vinculin A50I/β-catenin VBS ($\Delta G_{vinculin/\beta\text{-catenin M8P}}$ = −25.7 Kcal/mol, $p = 0.0050$—Supplementary Tables 8, 9). With vinculin/α-catenin(L344P), after 200ns the interaction of α-catenin(L344P) in S2 was lost and the contact in the highly charged segment in S1 decreased. Salt bridges and hydrogen bonds induced a partial movement of α-catenin(L344P) from the S1 groove together with a rearrangement in S2 due to intramolecular interactions in α-catenin(L344P) between E306/R308 and E310/R326 (Fig. 3i). The destabilization induced by L344P was confirmed by the free energy calculations ($\Delta G_{vinculin/\alpha\text{-catenin}}$ = −63.2 Kcal/mol vs $\Delta G_{vinculi/\alpha\text{-catenin(L344P)}}$ = −34.3 Kcal/mol $p < 0.0001$—Supplementary Tables 8, 9) and per-residue contribution (Fig. 3j). Taken together these results support the idea that different modes of interaction for the dimers formed between vinculin/β-catenin and vinculin/α-catenin are possible.

## The interaction between β-catenin and activated vinculin functionally supports force transmission at AJs

Due to structural similarity of β-catenin with well-characterized mechanotransduction binding partners of vinculin[52], we hypothesized that the vinculin/β-catenin interaction may also sustain physiological forces. To investigate this, we performed in vitro single-molecule magnetic tweezer experiments using a single-molecule detector containing β-catenin VBS and vinculin VD1 separated by for-min FH1 domain that functions as unstructured spacer (Fig. 4a, top and Supplementary Fig. 13a). At sufficiently low forces, it is expected that, if β-catenin VBS can effectively interact with VD1, a VBS–VD1 complex is formed (Fig. 4a, bottom). Such interaction should also be able to resist forces above a few piconewtons (pN) before dissociating resulting in a significant stepwise extension change (equivalent to the bead height change, ΔH) that can be observed in experiments (Fig. 4b). We conducted experiments using 26 independent tethers and observed dissociation of the β-catenin VBS–VD1 complex in eight cases (17 dissociation events from 8 individual tethers, exemplary experiments in Fig. 4b, inset and Supplementary Fig. 13). This reflects a relatively low (~ 30%) probability of complex formation, which reconciles with the calculated free energy from MM-PBSA analysis for vinculin/β-catenin complex (-28.9 Kcal/mol) as compared to that of the vinculin/α-catenin complex (−63.2 Kcal/mol). Within the reported dissociation events, we could observe a large variability in ΔH in response to forces ranging from 7 to 16 pN at a loading rate of 1 pN s⁻¹ as indicated by the counts of single events as a function of the applied force and step size (Fig. 4b, top and right histograms). Assuming a similar bending persistence of ~0.8 nm for the polypeptide polymer from VD1 unfolding and FH1 unlooping, we can predict the force-dependent step sizes resulting from VBS-VD1 dissociation and from VD1 subunit unfolding. Based on this calculation, three scenarios are possible: dissociation of VBS-VD1 without concurrent VD1 subunit unfolding (unlooping, theoretical ΔH ≈ 40−50 nm), dissociation with concurrent unfolding of one VD1 subunit (unlooping and partial unfolding, theoretical ΔH ≈ 60−90 nm), and finally dissociation with concurrent complete unfolding of VD1 (unlooping and unfolding, theoretical ΔH > 100 nm). In conclusion, these findings indicate that, once established, β-catenin VBS−VD1 complex can withstand physiologically-relevant forces (ranging from 7 to 16 pN), thereby facilitating a mechanically stable force transmission between β-catenin and the cytoskeleton network through association with vinculin.

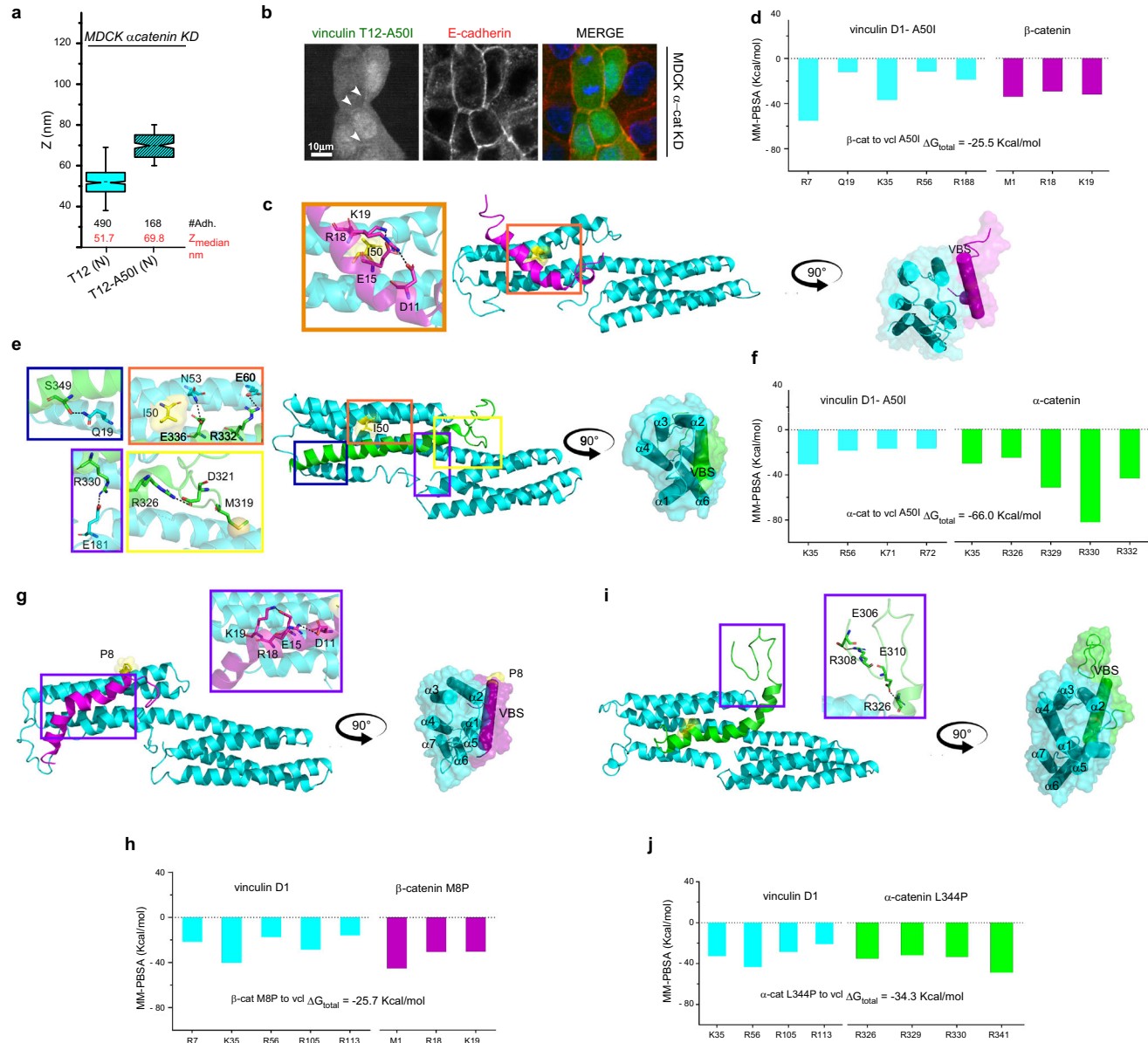

**Fig. 3 | Molecular dynamics analysis of vinculin and α- or β-catenin complexes with mutated sequences.** The n-terminus of activated vinculin with the point mutation A50I (vinculin T12-A50I) shifts higher up with respect to active vinculin without mutation. **a** Notched box and whisker plots indicating the median z-center position of the n-terminus of vinculin mutants, T12 and T12-A50I in MDCK α-catenin KD cells, obtained by SAIM (statical analysis reported in Supplementary Table 1). Box represents median, 1st, and 3rd quartiles; whiskers, 5th and 95th percentiles; #Adh. indicates the number of adhesions measured. **b** Localization of vinculin T12-A50I constructs to AJs in MDCK α-catenin KD in monolayer. Maximum intensity projections of vinculin (GFP, green channel) and E-cadherin (antibody probes, red channel), and merged image. Scale bar, 10 μm. **c** Cartoon representation of vinculin D1 mutated A50I (cyan)/ β-catenin (purple) dimer. The mutated Isoleucine amino acid is highlighted in yellow. The inset shows the main interactions and on the side the Lateral view of vinculin A50I/β-catenin dimer (90° rotation). **d** MM-PBSA per-residue energetic contribution for vinculin D1 mutated A50I/ β-catenin dimer. **e** Cartoon representation of vinculin D1 mutated A50I (cyan)/ α-catenin (green) dimer. The mutated Isoleucine amino acid is highlighted in yellow. The inset show the main interactions and on the side the Lateral view of vinculin A50I/α-catenin dimer (90° rotation). **f** MM-PBSA per-residue energetic contribution for vinculin D1 mutated A50I/ α-catenin dimer. **g** Cartoon representation of D1 vinculin wt (cyan)/ β-catenin (M8P) (purple) dimer. **h** MM-PBSA per-residue energetic contribution for vinculin D1/ β-catenin (M8P) dimer. **i** D1 vinculin (cyan)/ α-catenin (L344P) (green) dimer with the corresponding rotations. **j** MM-PBSA per-residue energetic contribution for vinculin D1/ α-catenin (L344P) dimer. The mutated amino acid in each mutant is highlighted in yellow in (**c**, **e**, **g**, **i**).

Furthermore, we assessed the tension within the cell-cell junctions of epithelial monolayer using a laser nanoscissor approach whereby elastic recoil after scission reports the tensional state of the junction[14,56,57]. In control MDCK wt cell grown in a monolayer, cell-cell junctions underwent rapid ballistic recoil upon laser ablation, followed by a more gradual recession (Fig. 4c–e and Supplementary Movie 7). We found that MDCK α-catenin KD displayed a significant reduction of the initial recoil rate indicative of a reduced junction tension relative to MDCK wt (1.34 μm/s ± 0.264 vs 1.77 μm/s ± 0.198, $p = 0.0030$).

However, the control tensional state was rescued in MDCK α-catenin KD expressing vinculin-T12 with recoil a recoil rate similar to MDCK wt (1.88 μm/s ± 0.451–Supplementary Tables 3, 4). These results show that in its open conformation (T12) vinculin is able to restore high junctional tension in a depleted α-catenin background, possibly due to a bypass mechanisms linking the actin cytoskeleton with E-cadherin via β-catenin. Then, we performed Fluorescence Recovery After Photobleaching (FRAP) experiments in MDCK epithelial monolayer to gauge changes in β-catenin diffusibility and turnover (Fig. 4f–h and

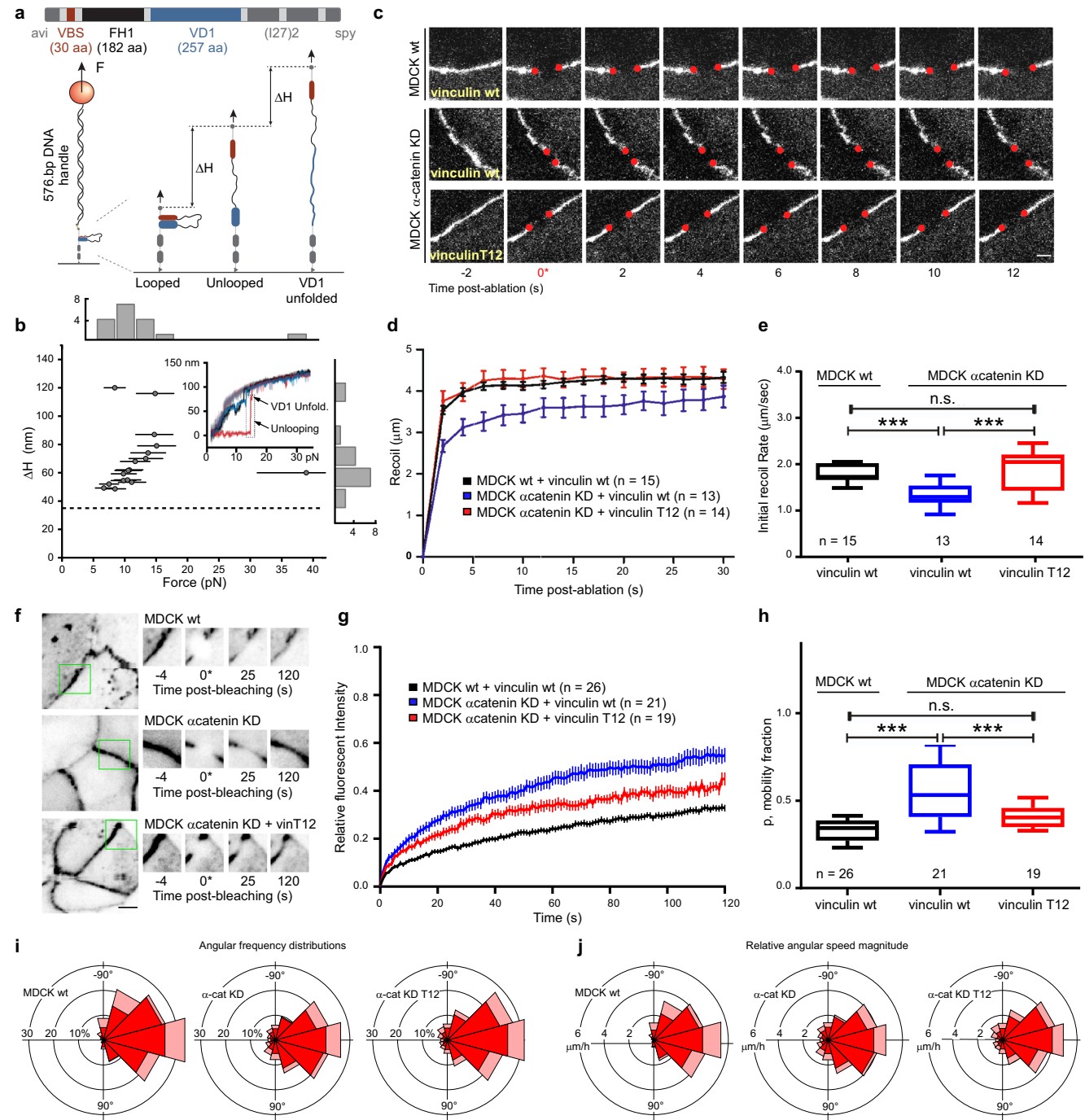

Supplementary Movie 8). In control MDCK cells, photobleaching was followed by a slow recovery with a half-life of 29.020 s ± 9.896 (Supplementary Tables 5, 6) and a small mobile fraction (0.331 ± 0.068), indicative of a relatively small dynamically exchanging fraction of β-catenin (Fig. 4g, h), which is consistent with the role of β-catenin in stabilizing E- cadherin in AJs[58,59]. Not surprisingly, depletion of α-catenin caused a significantly faster half-time (24.03 s ± 11.02) as well as an increased mobile fraction (0.553 ± 0.177). Similarly, vinculin-head construct, which lacks actin-binding site, displayed a high mobile fraction (0.598 ± 0.112) as well as a fast exchange (21.10 s ± 9.790), comparable to the MDCK α-catenin KD. However, upon the expression of vinculin-T12, we observed a partial recovery of the wildtype conditions with a low mobile fraction (0.402 ± 0.072) but a still fast recovery rate, similar to that of MDCK α-catenin KD alone (20.36 s ± 6.706). Finally, to further test force transmission between cells under these

conditions and evaluate their large-scale consequences, we probed for the ability of cells to maintain collective migration in a wound-model assay (Fig. 4i-j, Supplementary Table 7 and Supplementary Movies 9–11). Particle Image Velocimetry (PIV) analysis of these experiments, which vectorially represents the flow of cells, showed that expectedly MDCK wt cells migrate cohesively by coordinated collective migration (Supplementary Movie 9) and the preponderant frequency distribution (Fig. 4i, left) as well as the relative angular speed (Fig. 4j, left) being perpendicular to the migrating front (i.e., 0°). In contrast, chaotic motility and decreased coordination was observed in MDCK α-catenin KD as consequence of partial loss of force transmission between cells (Supplementary Movie 10) as indicated by the lower frequency distribution and relative angular speed perpendicular to the front of the tissue edge (i.e., 0°) (Fig. 4i, j, middle panels). Interestingly, Supplementary Movie 11 and right panels in Fig. 4i, j demonstrate that

**Fig. 4 | Junctional tension and stability are supported by activated vinculin interaction with β-catenin when α-catenin is depleted. a** Top, schematic representation of the construct used to conduct single protein manipulation by vertical magnetic tweezer to detect VBS-VD1 interaction and ability to functionally bear forces. The single molecule detector construct is composed of avi-tag, β-catenin VBS, formin FH1 domain, vinculin D1 domain, two repeats of titin I27 domain, and spy-tag. Bottom, schematic representation of the experimental design and possible outcomes. VBS-VD1 complex may form at low forces (looped). Upon application of pulling force to the magnetic bead by tweezer, dissociation events can be detected by measuring z position of the bead and resulting ΔH. Unlooping events correspond to instances of VBS-VD1 separation (unlooped). At higher forces, partial or complete VD1 unfolding events can be concomitantly observed. **b** 2D graph of the force-dependent steps sizes at a loading rate of 1 pN s$^{-1}$ illustrates the ability of β-catenin VBS to form a stable interaction with vinculin VD1 and sustain physiological forces ranging from 7 to 16 pN. 17dissociation events where beads height changes are significantly larger than 35 nm (horizontal dashed line−Supplementary Fig. 13) from 8 individual experiments are plotted. Large variability in ΔH observed results from complex behavior of FH1 unlooping (theoretical ΔH ≈ 50–60 nm) and partial (theoretical ΔH ≈ 60-90 nm) or full (theoretical ΔH > 100 nm) VD1 unfolding. Top and right histograms report the counts of single events as a function of the applied force and step size, respectively. The bars represent the relative uncertainty of force measurements generated by bead size variation and relative position of attachment of the tether on the bead. This has been estimated to be 20% of the measured force[90]. Inset, representative experiment shows a significant -120 nm step at -16 pN, indicating VBS-VD1 dissociation and VD1 unfolding. Three cycles of pulling at 1 pN s$^{-1}$ (dark lines) and release at 0.1 pN s$^{-1}$ (light lines) are shown with 1st cycle in black, 2nd cycle in red and 3rd cycle in blue. Experiments were performed at 37 ± 2 °C. **c** Representative time-lapse montages of recoil upon laser ablation. MDCK cells were co-transfected by the indicated constructs as well as ZO-1 to mark the cell-cell junctions. The edges of junctions are denoted by red dots. Scale Bar, 10 μm. **d** Trajectory of junctional recoil following ablation. Error bars indicate s.e.m. **e** Plot of initial recoil rate after laser ablation for the indicated conditions. Box represents median, 1st, and 3rd quartiles; whiskers, 5th and 95th percentiles. Statistical analysis in Supplementary Table 4, following ordinary one-way ANOVA with Sidak correction for multiple comparisons.*: p<0.05; ***: p < 0.001; n.s.: not significant. Representative video of recoil upon laser ablation for each condition indicated in (**d**, **e**) can be found in Supplementary Movie 7. Number of cell-cell junctions measured for each condition is indicated in the plots. Statistics and statistical analysis are reported in Supplementary Tables 3, 4. **f** Representative time-lapse images from Fluorescence Recovery After Photobleaching (FRAP) of β-catenin. (Right) Montage (zoom 2X) of the bleached regions (yellow squares in the main figures). Scale bar, 5 μm. **g, h** FRAP analysis of β-catenin for the indicated conditions. **g** Normalized recovery trajectory was fitted to a single exponential model, with mobile fraction of β-catenin relative to total. Error bars indicate s.e.m. **h** Plot of mobile fraction of β-catenin relative to total. Box represents median, 1st, and 3rd quartiles; whiskers, 5th and 95th percentiles. Statistical analysis in Supplementary Table 6, following ordinary one-way ANOVA without correction for multiple comparisons *: p < 0.05; ***: p < 0.001; n.s. not significant. Representative video of FRAP for each condition indicated in (**g, h**) can be found in Supplementary Movie 8. Number of cell-cell junctions measured for each condition is indicated in the plots. Statistics and statistical analysis are reported in Supplementary Tables 5, 6. **i, j** Analysis of directionality of collective migration suggests a partial rescue effect of activated vinculin (vinculin T12) on α-catenin depletion. Rose plots show the angular frequency distribution of the integrated velocity (expressed as percentage of the total−**i**) and the relative angular speed magnitude (μm/hour−**j**) of migrating MDCK wt, MDCK α- catenin KD, MDCK α-catenin KD transfected with vinculin T12 cells. Red indicates the mean and pink indicates standard deviations. Representative video of migration assay for each condition can be found in Supplementary Movies 9–11 and statistical analysis in Supplementary Table 7.

expression of T12 in α-catenin KD cells was able to largely rescue the cohesive migration phenotype seen in wildtype cells. It must be noted that by experimental design, contact inhibited cells, such as MDCK cells, will be prevented from migrating toward the bulk of the tissue and thus a preferential motion perpendicular to edge of the tissue (along the x-axis) is to be expected. As result, motion is directionally skewed, and statistical analysis has reduced degrees of freedom. Nonetheless, statistical analysis indicates high correspondence between mode of migration of MDCK wt and MDCK α-catenin KD transfected with vinculin T12 mutant ($p$ = 0.81 and 0.97 for the angular frequency and mean speed magnitude, respectively−Supplementary Table 7). Concomitantly, low similarity was observed when comparing MDCK α- catenin KD with the other samples (0.17 < $p$ < 0.37 for all samples when compared to α-catenin KD). Importantly, equal amounts of vinculin wt and vinculin T12 in MDCK wt and MDCK α-catenin KD were measured at the AJ (Supplementary Fig. 5b, c), thus proving that results from laser ablation, FRAP, and wound-model assay should be interpreted as consequence of the type of vinculin configuration/ mutation and not to the relative amount of protein at the AJ.

Taken together, these results suggest that a functional link between the open conformation of vinculin and β-catenin can mediate force transmission between the actin cytoskeleton and cadherin when α-catenin is depleted. For instance, such function could potentially explain how PC3 cells, which are derived from prostate cancer, are capable of migrating collectively (Supplementary Movie 12) despite lacking α-catenin (expression levels in Supplementary Fig. 4c, d). However, this observation remains only suggestive of possible conditions where force transmission by the β-catenin/vinculin complex can possibly explain yet unresolved pathological scenarios.

### In the presence of α-catenin, β-catenin can stabilize the adhesion by binding to S3 on vinculin

Besides providing an alternative path to maintain function AJ even in low or absent α-catenin, our findings open the fascinating possibility that yet-undiscovered configurations of the catenins/vinculin complex

centered on vinculin. Thus, we tested the hypothesis that α-catenin and β-catenin could simultaneously interact with open conformation of vinculin to reinforce and stabilize the cell-cell adhesion. As our finding demonstrated that β-catenin can interact with vinculin open conformation, we propose that α-catenin initially priming of vinculin is necessary. Thus, we generated a ternary complex starting from the vinculin/α-catenin VBS dimer (i.e., last frame of the simulation, Fig. 1e) and performed the docking of the β-catenin on the dimer vinculin/α-catenin without any restriction or constriction.

The docking revealed that the most energetically favorable binding for β-catenin VBS was on S3 in vinculin in an antiparallel orientation to α-catenin VBS positioned on S1 and S2. S3 is formed by α-helix 1 and 4 and it is composed mainly of nonpolar residues. S3 is described as one of the regulatory sites for vinculin activation, being involved in the interaction with vinculin tail to allow for the head-to-tail closed conformation. Indeed, vinculin mutation T12 that we have discussed in the previous sections, contains four mutated residues in the Vt (D974A, K975A, R976A, and R978A) which reduce the affinity of head to tail[53,54]. To analyze the stability of β-catenin VBS in S3 over time, we performed MD under the same conditions used to analyze the dimers (Figs. 2, 3). The MD analysis (Supplementary Movies 13, 14) showed that during the first 450 ns, β-catenin VBS maintained an α-helix conformation. Nevertheless, at 450 ns, the C-terminal of β-catenin VBS partially unfolded and moved into the groove between α-helix 1 and α-helix 4 of vinculin (Supplementary Fig. 14 and Supplementary Movie 13). This could suggest that when S1 is occupied by the presence of another interacting partner (i.e., α-catenin in this case), which primes vinculin in an opened conformation, β-catenin can interact with S3, thus interfering with head-to-tail interaction and locking vinculin in its open conformation. MD analysis for α-catenin VBS (Supplementary Movie 14) showed that during the first 180 ns, α-catenin VBS maintained a stable interaction in S1 and S2 in vinculin mainly through the conservation of hydrogen bonds and salt bridges (the pairs N53/E336, E60/R332, E60/R329, and E66/R326) and hydrophobic interactions (residues L318, I313, M319, and L312). However, the

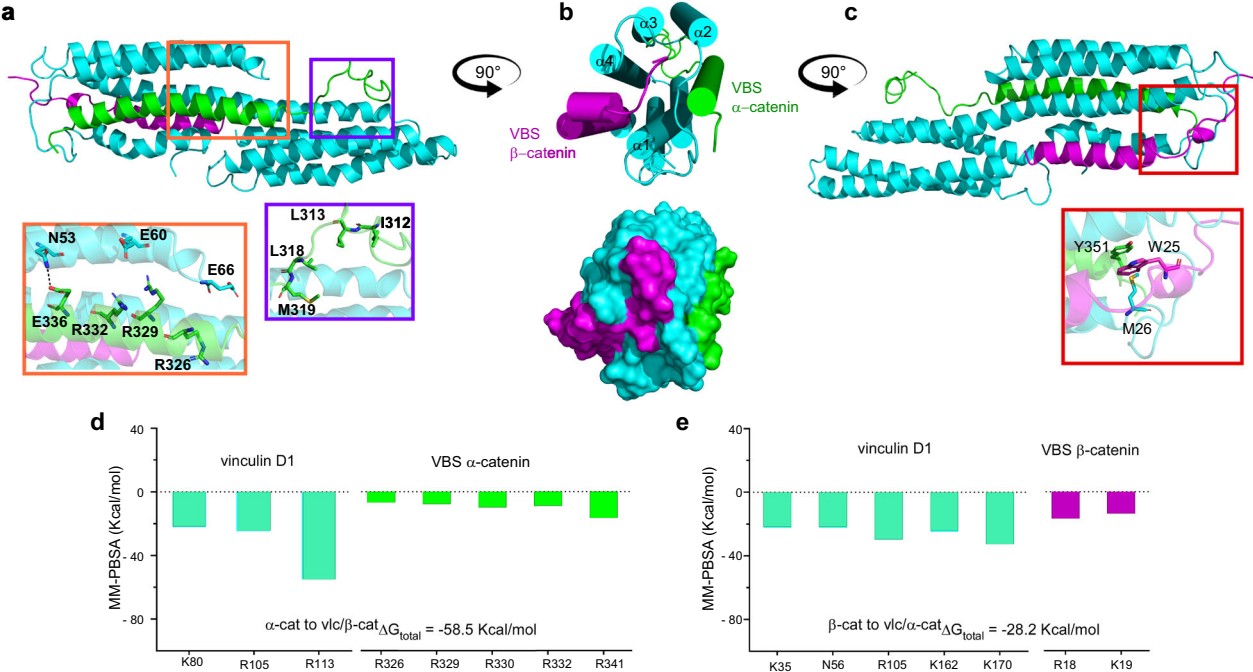

**Fig. 5 | MD analysis of putative vinculin - α-catenin - β-catenin trimeric complex.** Cartoon type representation of the trimer vinculin D1 domain (cyan)/α-catenin VBS (green)/ β-catenin VBS (purple). **a** Focus on vinculin D1 domain (cyan)/ α-catenin VBS (green) in S1 binding pocket. Insets show the intra and inter-molecular residue interaction. **b** The complex is rotated 90° to show a cartoon and surface representation of lateral view of the trimer α-catenin/ vinculin/ β-catenin showing both catenin peptides coupled to vinculin D1. **c** Focus on vinculin D1 domain (cyan)/ β-catenin VBS (purple) in S3 binding site. The inset shows the new interaction generated by residue W25 that provides a key pi-stacking interaction with Y351 and establish the vinculin/α-catenin/β-catenin trimer. MM-PBSA per-residue energetic contribution for (**d**) vinculin D1/α-catenin VBS and (**e**) vinculin D1/ β-catenin VBS to the trimer stability.

rotation of the α-catenin VBS on S1 at 500–600 ns destabilized the interaction in S2 causing a partial loss of the α-helix structure of α-catenin. This was due to the increased distance between the residue R326 of α-catenin VBS and E60 of vinculin, which moved from 3 Å to a distance greater than 20 Å due to the loss of salt bridges, thus causing the destabilization (Fig. 5a–c and Supplementary Fig. 15).

These results support the putative generation of a vinculin/α-catenin/β-catenin tertiary complex in which vinculin plays a pivotal role interacting simultaneously with both catenins with β-catenin VBS sitting in S3 ($\Delta G_{\text{vinculin-α-catenin /β-catenin}} = -28.2$ Kcal/mol) and α-catenin in S1 ($\Delta G_{\text{vinculin- β-catenin /α-catenin}} = -58.5$ Kcal/mol) to lock vinculin in an extended conformation (Fig. 5d, e).

## Discussion

The current view of mechanotransduction at cadherin-mediated adhesions places α-catenin as the principal force-transducing protein in the adhesion complex, serving as the dominant mechanical link between the cadherin signaling layer at the membrane and the actin cytoskeleton[60]. While this view has been recently challenged[35], our results, based on direct measurement of tensional state of AJ, clearly demonstrate a significant loss of tension in α-catenin KD cells (Fig. 4d, e), confirming the central role of α-catenin under physiological conditions. However, growing body of evidence suggests that alternative pathways could mediate force transmission at the AJ[29–35]. Here, we demonstrate that β-catenin can form a physical connection with open vinculin, capable of rescuing force transmission in absence of α-catenin. This opens two interesting scenarios: one that explains how cadherin mediated adhesion could be maintained when α-catenin is reduced or absent, as seen in a variety of cancer cells (Fig. 6, dimers/pairs), and the second where β-catenin prevents vinculin to switch into its head-to-tail autoinhibited conformation, thus providing a mechanism to reinforce AJs (Fig. 6, trimer).

To date, the interaction between vinculin and β-catenin has been relatively under-studied at the biochemical and biophysical levels as it has classically been presumed to merely connect cadherins and α-catenin. However, it has been reported that different types of carcinomas are able to migrate collectively (Supplementary Movie 12 and Supplementary Fig. 4c, d) despite having low or no α-catenin expression[37,38,61,62]. Our results clearly demonstrate that, in absence of α-catenin, a functional direct interaction between β-catenin and non-autoinhibited vinculin can form to sustain physiologically relevant forces (Fig. 4b). A loading rate in the order of pN s$^{-1}$ is anticipated by force-transmission to membrane receptors that orientates the corresponding force-transmission supramolecular linkage (as discussed in ref. 63) and mechanical unfolding of force-bearing domains (such as vinculin domains[64]) due to actomyosin contraction. The observed forces of 7 to 16 pN for the disruption of the β-catenin VBS-VD1 complex are in the similar range where mechanical unfolding of vinculin domains occurs[64]. Therefore, this complex, once formed, is able to support force transmission that could lead to a downstream cascade of mechanotransduction events mediated by vinculin.

While the nature of this interaction still remains to be fully understood from an experimental point of view, our molecular modeling suggests that β-catenin could form stable interaction with the S1 portion within the 4-helices pocket of vinculin. The mechanism by which vinculin could be activated in order to expose S1 for β-catenin engagement in low or absent α-catenin is still an open question. Our previous studies demonstrated that vinculin may assume metastable conformations induced through phosphorylation or tension causing a plethora of intermediate conformations between a close autoinhibited and fully extended form[14]. Importantly, it has been previously reported that PC3 cell aggregates are maintained by N-cadherin adhesions[37], and we have shown that in N-cadherin adhesion complexes, vinculin is in an extended conformation[14]. This could allow for the interaction with β-

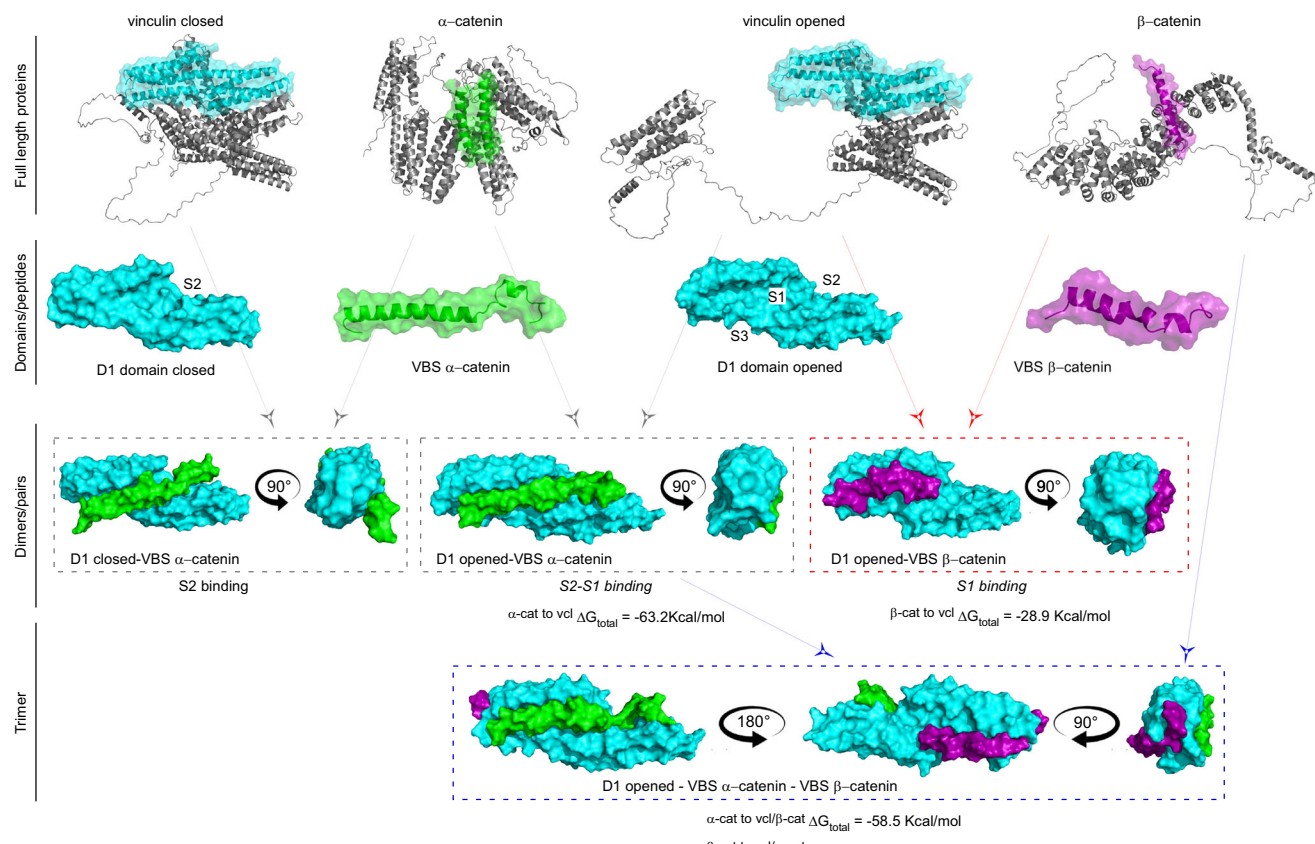

**Fig. 6 | Different conformational arrangements of vinculin-catenin complex.**
Cartoon representations of full-length proteins of the mechanotransduction module (vinculin, α-catenin, and β-catenin). Vinculin is represented in its closed and open conformation. In colors, we highlighted the putative peptides/domains involved in the interactions. In the second row (Domains/Peptides), the interacting regions highlighted above (vinculin D1, VBS α-catenin, and VBS β-catenin) are individually visualized as surface representations. In vinculin D1 domain, the sites S1, S2, and S3 are indicated. In the third row (Dimers/pairs), interactions between couples of Domains/peptides are shown. In the last row, the proposed new trimer of vinculin/β- catenin/α-catenin is shown, with vinculin as central molecule. For dimers and the trimer, the calculated MM-PBSA per-residue energetic contribution are indicated.

catenin. Future experiments will be necessary to fully address the interplay of these connections in regulating AJ properties.

Our data also suggest that β-catenin could play a role in stabilizing and reinforcing the AJ by directly interacting with vinculin in the presence of α-catenin. Indeed, our MD simulations suggest that β-catenin could directly interact with S3 domain of vinculin to form a ternary complex with α-catenin in S1 and S2 when vinculin is in its non-autoinhibited form. In our view, this could be achieved by a multistep mechanism where the assembly of the AJs components occurs sequentially and, unlike the on/off system predicted by the current model[22,65], graded force responses can be obtained via subsequent conformational rearrangements within the vinculin/catenins trimer[22,62]. In the first step, the ternary complex of E-cadherin/β-catenin/α-catenin may serve as the initial nucleation sites for cell-cell adhesions. Since α-catenin unfolds under a relatively low ~5 pN tension[22], this may simultaneously recruit vinculin and disrupt vinculin autoinhibition. Interestingly, a recent study shows that mechanical exposure of one VBS in talin is sufficient to activate full-length vinculin and engage in a high affinity interaction[66]. Thus, it is not farfetched to consider that α-catenin could activate vinculin via a similar mechanism. Subsequently vinculin may reinforce the connection to the actin cytoskeleton through both the actin-binding tail domain (Vt) and via the association of the actin elongation factor VASP to the poly-proline-rich linker region[50]. In this same direction, it has been reported the dependent geometry-cooperative effect for the reinforcement of the intercellular contacts to the formation of quaternary complexes of E-cadherin/β-catenin/α-catenin/vinculin

when applying tension in the orientation towards the (-) end of actin filaments[67]. As depicted in Fig. 6, once activated and fully extended, vinculin D1 could become engaged with the N-terminal site on β-catenin to provide direct linkage of vinculin with the E-cadherin/β-catenin complex. Interestingly, this interaction with S3 domain may also lock vinculin in an open conformation by impeding the head to tail interaction, thus stabilizing the adhesion. As shown in junctional tension measurements (Fig. 4), the activated vinculin/β-catenin interaction could be force-bearing, supporting a comparable magnitude of tension as in the control. Since α-catenin has been demonstrated to exhibit a catch-bond behavior with peak lifetime at <10 pN when binding to F-actin[16], the bond lifetime is expected to be diminished at higher tensional states, such as those achieved when vinculin is recruited at the AJ. We speculate that this may favor the disengagement of the activated vinculin from α-catenin. Since the N-terminal vinculin binding site of β-catenin is in proximity and does not overlap with α-catenin/β-catenin binding site[23], it is conceivable that vinculin head domain could translocate to link up with β-catenin directly. In other words, α-catenin could be considered as a catalyst that activates and potentiates vinculin. Following such activation, the mechanically primed vinculin could translocate to serve as a mechanical linker on β-catenin as described in this study, notwithstanding that potentially other partners such as α-actinin or Mena/VASP may be involved in alternative mechanotransduction pathways[68,69]. Interestingly, an analogous vinculin translocation process has recently been described in integrin-based focal adhesions[70]. Further investigation is needed to verify whether a

potential comparable molecular design principle may be at play at different mechanical interfaces of the cells.

In conclusion, deciphering how proteins of the adhesive complex self-organize and integrate into functional higher-order machineries that allow cells to aggregate into mechanically resistant and highly dynamic tissues, is the first step to provide a mechanistic understanding of the mechanical basis for development and cancer transformation and for the generation of predictive tools to engineer tissues for regenerative medicine.

## Methods

### Protein structures

Full-length structures of the proteins analyzed in this study have been obtained from AlphaFold database[40,41]. Vinculin (UniProt ID P18206), α-catenin (UniProt ID P35221) and β-catenin (UniProt ID P35222) have been used only for graphical purpose for the representation of the position of the domains (vinculin D1, α-catenin VBS and β-catenin VBS) studied by computational modeling.

The crystal structure of the vinculin D1 domain in complex with VBS α-catenin segment reported in the Protein Data Bank database (PDB code: 4EHP[23]) was used. The acetate ions and water molecules were removed from the PDB file. The VBS α-catenin segment (305 to 355) was conserved.

To perform an amino acid conservation analysis the VBS segments of the Talin VBS1 (PLLQAAKGLAGAVSELLRSAQPASAEP), VBS2 (KLLS AAKILADATAKMVEAAKG), and VBS3 (ELIECARRVSEKVSHVLAALQA), α-actinin (LIQNEVENITRAITTLLQEWGV), α-catenin (DDRRERIVAECNA VRQALQDLL), and β-catenin (MATQADLMELDMAMEPDRKAAVSHWQ QQSY) were selected from canonical sequence of Uniprot database[71]. The alignment was performed with MUSCLE server[72], available on EBItools web[73], using default parameters and the analysis and visualization of the conserved aminoacid was performed with the Jalview v2.11.2.26 software here the conserved amino acid was highlighted blue[74]. It has to be noted that, as opposed to the rest of the VBS segments, the VBS sequence of α-actinin is inverted as previously reported[51].

In order to generate a three-dimensional structure of N-terminal segment of β-catenin, an ab-initio model using Robetta software was obtained. It is important to mention that both, secondary structure prediction (using PSIPRED, http://bioinf.cs.ucl.ac.uk/psipred/) and the results reported by Peng and collaborators[55], describe an α-helix for the N-terminal segment of β-catenin. Thus, based on the vinculin/α-catenin complex in PDB database an α-helix was obtained for β-catenin which was used to superimpose on the αcatenin coordinates to generate a complex vinculin/β-catenin. In addition, the mutant complexes; vinculin A50I, α-catenin L344P, and β-catenin M8P were obtained using Mutagenesis wizard in PyMOL2 software[75].

The trimer complex vinculin/α-catenin/β-catenin was generated using Cluspro2.0[76,77] First, the complex of α-catenin/vinculin was used as "receptor-like" using the coordinates of the last frame of the molecular dynamic, while the N-terminal VBS of β-catenin was used like "ligand". The docking was made without restrain or constrain. Finally, ClusPro2.0 provides the best energetic and probabilistic protein-protein complex. Therefore, based on the score and cluster size the best trimer model was selected. Each structure was corrected in order to adjust the protonation state and the hydrogens missing using Amber LEaP script[78] to obtain protein models considering the physiological condition.

### Molecular dynamics (MD)

MD simulations were performed for all the proposed complexes using Amber program and AMBER99SB force field[78,79], where a cutoff of 8 Å was used for nonbonded interactions, and long-range electrostatic interactions were treated with the particle mesh Ewald (PME) method. Each system was inserted in an explicit water box using transferable

intermolecular potential with 3 points (TIP3P) as water model. All simulations were carried out under periodic boundary condition and neutralized by adding necessary $Na^+$ ions. The equilibration was carried out following 6 steps going from 100 to 300° K. Each system was run for 1 μs total without restraints and using a time step of 0.002 ps in the NPT ensemble (considering the Berendsen barostat set at 1 by default). The analysis of the trajectories of each system and APBS calculations was performed using the VMD[80] and PyMOL2 software[75].

The molecular mechanics Poisson–Boltzmann surface area (MM-1PBSA)[81] approach was used to estimate the binding affinity between vinculin/α-catenin, vinculin/β-catenin, vinculin(A50I)/α-catenin, vinculin(A50I)/β-catenin, vinculin/α-catenin(L344P), vinculin/β-catenin(M8P), vinculin/α-catenin/β-catenin; for this were used the most stable and representative snapshots, with a time step of 10 ns for a total of 1 μs of trajectory. Additionally, the energy decomposition of each residue was calculated, and the extraction of the total energy contribution of selected residues (based on the main interactions) was plotted.

The script for MM-PBSA was taken from[81] and it is here reported:

```
¨input for running PB
&general
startframe = 0,  endframe = 10000,  keep_file = 2,  interval = 100,
verbose = 1,
/
&gb
igb = 2, saltcon = 0.15
/
&pb
inp = 2, istrng = 0.150, exdi = 80.00, indi = 4.0,
fillratio = 4.0, radiopt = 0,
/
&decomp
idecomp = 1, dec_verbose = 0¨
```

### Cell culture and transfection

MDCK (Madin-Darby Canine Kidney) and MDCK α-catenin KD cells were a kind gift from W. James Nelson−Stanford University. MDCK cell line with stable expression of α-catenin shRNA have been created in Dr. Nelson´s lab as reported in ref. 82. They are maintained with G418 (1 mg/ml) for a minimum of 9 days before use to obtain about 90% knock down of α-catenin expression level (Supplementary Fig. 4b). MDCK α-catenin KO cell lines were also generated from parental MDCK from W. James Nelson group. These cell lines were cultured in DMEM media with high glucose, supplemented with 10% Fetal Bovine Serum (FBS), and 100 units/mL of Penicillin/ Streptomycin (Life Technologies Carlsbad, CA). Cells were transfected with ~10 μg of plasmid DNA encoding the protein of interest by electroporation using Neon® Transfection System (Life Technologies). The following settings were used for the electroporation: one pulse with 20 ms width at a voltage of 1650 v for ~1 × 10⁶ cells.

PC3 (Prostate Cancer) cells were a kind gift of Dr. Alejandro Godoy −Universidad San Sebastian (Chile). They were cultured in RPMI media supplemented with 10% Fetal Bovine Serum (FBS), and 100 units/mL of Penicillin/Streptomycin (Life Technologies Carlsbad, CA). No cell lines used in this study were found in the database of commonly misidentified cell lines that is maintained by ICLAC and NCBI Biosample (https://iclac.org/databases/ncbi-biosample/). The cell lines used were monthly tested for mycoplasma contamination by PCR method.

### Generation of α-catenin KO MDCK cell line

Biallelic knock-out of α-catenin in MDCK cells were generated by CRISPR/Cas9-mediated genomic editing using dual guide RNAs to induce exon skipping[83]. The dual-guide RNA oligonucleotides used is designed using the CHOPCHOP web tool (https://chopchop.cbu.uib.no) and synthesized by Integrated DNA Technologies. Each guide

sgRNA sequence is designed to flank exon2 of *Canis familiaris* CTNNA1 gene, with sequences: sgRNA-A (5′-TAGGGGCTCCGGTAT CGATGGGG-3′) and sgRNA-B (5′-TCTGGCAGTTGAAAGACTGTT GG -3′), respectively. The oligo sequences were separately ligated into the pSpCas9(BB)-2A-Puro (PX459) V2.0 backbone plasmid (Feng Zhang laboratory, Addgene plasmid # 62988), which contains Cas9 gene with BbsI cloning site under the control of U6 promoter. The hU6-F primer (5′-GAGGGCCTATTTCCCATGATT-3′) was used for sequencing to confirm the presence of sgRNA oligos after cloning (Axil Scientific Sequencing). MDCK cells ($5 \times 10^5$ cells per transfection) was electroporated with 5 μg of plasmids containing either sgRNA-A or sgRNA-B, by utilizing a Neon™ Transfection System (100 μL Kit, Thermo Fisher, Cat. No. MPK10096) with pulse rate of 1650 V, 20 ms pulse width and 1 pulse. The transfected cells were grown for 48 h before selection using 2 μg/mL puromycin (Sigma-Aldrich, Cat. No. P8833). Following puromycin selection, cells were suspended and cultured in 96 wells with 2–3 starting cells per well. Genomic DNA was then extracted for each well and PCR was performed to detect deletion band and non-deletion band, using primers flanking the CRISPR cut sites and primers in the excised region, respectively. Biallelic deletion single-cell clones are selected from wells which exhibit deletion bands and no non-deletion bands. Upon cell growth, DNA extraction and PCR was performed to confirm biallelic deletion. Western blotting and immunofluorescence microscopy were then performed to confirm the absence of α-catenin expression (Supplementary Fig. 6).

### Fluorescent protein fusion constructs

FP fusion construct of vinculin was created in the laboratory of Michael W. Davidson, The Florida State University, and available from Addgene repository. EGFP-vinculin head (residue 1-258) and the N-terminal fusion of vinculin-T12 were obtained from Addgene (#46270, #46266, contributed by Susan Craig, Johns Hopkins University). The C-terminal fusion of vinculin-T12 mutant of vinculin were generated by Hongying Chen (MBI Protein Expression Facility) as indicated in ref. 14. All constructs created in-house were verified by sequencing. Localization to the cell-cell junctions was verified in MDCK cells plated on fibronectin-coated coverglass prepared by incubating 10 μg/mL of bovine fibronectin (F1141, Sigma) in PBS for 1 h at 37 °C in humidified atmosphere, cultured to confluence, and imaged by epifluorescence microscopy (Supplementary Figures in Bertocchi et al, NCB 2017[14]).

### Surface modification of imaging substrates with oriented cadherin extracellular domains

Silicon oxide wafers, p-type (100)-orientation with ~500 nm thermal oxide (Bondatek; Addison Engineering, CA) were measured for their thermal oxide thickness with nanometer-precision using an ellipsometer (UV-VIS-VASE, JA-Woollam) at SMART (Singapore-MIT alliance for Research and Technology, Singapore).

Wafers were then pre-treated as described in ref. 10. Briefly they were cut into ~1.2 cm × 1.2 cm chips with a diamond-tip pen, pre-rinsed with distilled water, sonicated in 100% Acetone, and then in 1 M potassium hydroxide for 20 min each, after a wash in distilled water. Silicon wafers were silanized for protein conjugation by incubation with 3-glycidoxy-propyl-dimethoxy-methylsilane (Sigma) (0.045% in 100% EtOH) for 1 h, then cured at 110 °C for 1 h. Silanized substrates were washed with 70% EtOH first and then distilled water and air-dried with nitrogen. Subsequently, the substrates were incubated with goat anti-human $F_c$ fragment specific antibody for E-cadherin substrate (Jackson ImmunoResearch #115-005-008), at 1 μg/cm² in 0.1 M pH 8 borate buffer and incubated at 4 °C overnight in a humidity chamber. After rinsing with PBS, the substrates were neutralized by aminoethoxy-EtOH (Sigma) in NaHCO₃ (100 mM, pH 8.3) for 1 h. Following a washing step, the substrates were incubated for 2 h with human E-cadherin-$F_c$ (R&D system 648-EC-100) at 1 μg/cm², rinsed with

PBS, and blocked with 0.2% pluronic acid (Sigma) in PBS for 20 min at room temperature. All washing steps were done in PBS with $Ca^{2+}$ and $Mg^{2+}$ to avoid collapse of the cadherin structure.

Transfected cells were detached by diluted trypsin (1:5 in PBS) and replated in serum free medium onto the silicon wafers, pre-coated with E-cadherin-$F_c$ (as described above) and incubated at 37 °C in 5% $CO_2$.

### Nanoscale-precision Z-position measurement by surface-generated structured illumination

SAIM[14,45–48] is a technique that allows measurement of the distance of fluorescent dyes to the surface with nm resolution. Its principle is very simple. The sample, containing a fluorophore fused to a protein of interest, is placed on a silicon wafer that acts as a mirror thanks to the presence of an oxide spacer. The fluorophore is excited by a standing wave that results from the interference between the direct and the reflected beams of light and, whose pattern varies as a function of the incident angle of the projected beam. In turn, a unique fingerprint of emitted intensity can be obtained as a function of incident angle of the beam depending on the distance to the surface (z-position). Thus, by fitting the measured intensity curve (as function of the incident angle) with the predicted output from the standing wave model, it is possible to calculate the z-position of each fluorophore, independent of their intensity.

Briefly, SAIM imaging was performed on a Nikon Eclipse Ti inverted microscope (Nikon Instruments, Japan), equipped with a motorized TIRF illuminator, using a 60 X ApoTIRF objective lens with NA 1.49, a sCMOS camera (Orca Flash 4.0, Hamamatsu, Japan), and a laser combiner (488 nm and 561 nm, Omicron Laserage, Germany) coupled with a polarization-maintaining optical fiber. Alternatively, we perfomed the experiments on a Nikon Eclipse Ti2E inverted microscope (Nikon Instruments, Japan), equipped with a motorized TIRF illuminator, with a 60 X ApoTIRF objective lens with NA 1.49, a EMCCD camera (Andor iXon 897).

Cells on the biomimetic E-cadherin substrate were fixed for imaging ~4–6 h after replating, using 4% paraformaldehyde in PHEM buffer (PIPES 60 mM, HEPES 25 mM, MgCl2 2 mM, without EGTA, pH 7.0). For imaging, silicon wafers were placed into a PHEM-filled 27-mm glass-bottom dish (Iwaki, Japan), with the sample side facing downward, maintained at neutral buoyancy by a thumb screw.

Fluorescence images were acquired between 0° (normal) and 52° with 4° incidence angle interval, using pre-tabulated values for either 488 nm (for eGFP) or 561 nm laser (for photo-converted EosFP). As needed, photoconversion of tdEos2 was carried out using LED excitation and DAPI filter set (Lumencor SOLA, Beaverton, OR). Analysis was performed using a custom written IDL-based software, as indicated in ref. 14. Binary masks for region of interest (ROI) to analyze were defined by simple thresholding of background subtracted images. Topographic height (z) and other fit parameters were determined by fitting the theoretical angle-dependence curve to experimental angle-dependence intensity by the Levenberg–Marquadt nonlinear least square method, for each pixel in the ROI. The median z (height)-position value of each ROI was denoted $z_{center}$ and used as the representative z-position for a given ROI. Topographic Z map were plotted using color to encode z-position.

### Three-dimensional superresolution microscopy by interferometric photoactivated localization microscopy (iPALM)

For actin imaging, a specific buffer for cytoskeleton preservation was used (CB: 10 mM MES, pH 6.1, 150 mM NaCl, 5 mM glucose, and 5 mM MgCl2) was used. Cells were briefly washed with warm (37 °C) calcium-containing CB, and pre-fixed by 0.2% glutaraldehyde in CB for 2 min, followed by 2% glutaraldehyde 0.1% Triton-X100 in CB for 10 min. Afterward, quenching of glutaraldehyde autofluorescence was obtained by incubation for 7 min in freshly prepared 0.1% NaBH4 in CB, followed by 3 rinses in CB for 10 min each. Samples were then

incubated in a humid chamber with 0.33 μM of AlexaFluor 647 phalloidin (Life Technologies) in PHEM (without EGTA) buffer for either 30 min at room temperature or overnight at 4°C. 100 mM cysteamine in PHEM buffer was used as imaging buffer to promote blinking. Samples were sealed by epoxy and vaseline-paraffin mixture and mounted on a custom-machined stainless-steel holder equipped with dual piezo actuators (Physik Instruments).

For imaging, we used an iPALM 3D superresolution microscope system that was built, calibrated, and operated as described earlier[14]. Briefly, a pair of dual-opposed 60X NA 1.49 Apo TIRF (Nikon Instruments) objective lens were aligned and focused using Picomotors piezoelectric actuators (Newport, Irvine, CA). The custom-manufactured 3-way Hess multiphase beamsplitter (Rocky Mountain Instruments, Lafayette, CO) was positioned by a 5-axis Picomotor actuators (Newport). Optimal mutual phase interference was adjusted by a dielectric mirror on a z-tip-tilt piezoelectric mount (Physik Instrument, Germany), index-matched to the beamsplitter. All major optical parts are mounted on custom-machined Invar parts to minimize drift. Three deep-cooled back-illuminated EMCCD cameras (Ixon Ultra, Andor, UK) were used for acquisition of raw images in frame transfer mode (50 ms exposure time), each controlled by a dedicated computer. Room temperature control was overridden prior to experiments by a custom-installed switch and allowed to equilibrate for minimal temperature fluctuation[14].

Calibration, alignment, and acquisition were performed using custom-written codes in LabVIEW (National Instruments), while image processing and data analysis were performed custom-written IDL software[14]. Z-coordinates were determined from a calibration curve recorded pre-acquisition for each imaging site by scanning the piezo sample holder in 8 nm step. Channel registration and drift correction was performed using plasmonic fiducials immobilized in coverglasses, while superresolution images were reconstructed using hue-encoding for z coordinate and normalized Gaussian.

## Immunofluorescence microscopy

Cells were fixed using 4% paraformaldehyde in PHEM buffer for 12–15 min, followed by 0.1% Triton-X100 in PHEM buffer for 3 min at RT, and blocking with 4–5% BSA for 1h at RT. Primary antibodies were diluted in PHEM buffer with BSA 3–4% for 1h at RT. The following antibodies have been utilized at the indicated dilutions: Rat anti E-cadherin (DECMA) antibody (Abcam #ab11512) was used at a dilution of 1:1600. Rabbit anti-β-catenin (Abcam #ab2365) was used at a dilution 1:200. Mouse anti-vinculin (Sigma #V9131) at 1:400. Secondary antibodies Alexa Fluor (Abcam) against the specific species were diluted 1:400 in PHEM buffer 3% BSA. Samples were hard-mounted using DAKO mounting media.

Confocal imaging of immunostained samples was performed using a spinning-disc confocal microscope based on a CSU-W1 Spinning Disk (Yokogawa) with a single, 70-μm pinhole disk and quad dichroic mirror along with an electron-multiplying charge-coupled device (EMCCD) (Princeton Instruments, ProEM HS 1024BX3 megapixels with 30-MHz cascade and eXcelon3 coating) using a Plan Apo 60x NA 1.20 or 100x NA 1.45 objective lens. Line profile was generated using FIJI plot line profile functions, and raw gray values were normalized to each channel, and plot in GraphPad.

## Western blot

MDCK WT, PC3, and MDCK alpha-catenin KD cells or MDCK alpha-catenin KO were lysed in RIPA lysis buffer (50 mM Tris-HCl pH 7.4, 150 mM NaCl, 1% IGEPAL), supplemented with Protease Inhibitor Cocktail Set III (Merck, United States). Total proteins were quantified using BCA protein assay kit (Thermo Fisher Scientific, USA). 10 μg of total protein was loaded into 10% SDS-Polyacrylamide gel electrophoresis (PAGE) gels and transferred to polyvinylidene difluoride (PVDF) membranes (Thermo Fisher Scientific, USA). Membranes were blocked with 5%

nonfat dry milk (wt/vol) in TBS-T buffer (20 mM Tris, pH 7.4; 100 mM NaCl; 0,5% tween-20). Membranes were incubated with the following primary antibodies and their respective dilutions: rabbit monoclonal anti-alpha-catenin antibody (Abcam #ab51032), at 1:3000; rat monoclonal anti-tubulin antibody (Abcam, #18251), at 1:4000; mouse monoclonal anti-Vinculin antibody (Millipore Sigma #V9131), at 1:3000. Anti-mouse IgG,and anti-rabbit IgG (Invitrogen, Thermo Fisher Scientific #31430 and #31460 respectively) anti-rat IgG (Cell Signaling, #7077) Horseradish Peroxidase (HRP) conjugated secondary antibodies. All secondary antibodies were used at 1:5000 dilution. HRP activity was detected with Sun and SuperNova substrates (Cyanagen, Italy). Western blot densitometric quantification was performed using the ImageJ software.

## Co-IP analysis

Cell lysis, protein extraction, and co-IP have been performed following the protocols indicated in ref. 84 with slight modification. Briefly, MDCK and MDCK a-catenin KD cells transfected with different vinculin constructs (eGFP vinculin wt, vinculin T12 eGFP, vinculin T12 A50I eGFP, vinculin head eGFP, vinculin Y822E eGFP and Y822F eGFP) were washed twice in HS buffer (20 mM Hepes, pH 7.4, and150 mM NaCl), and lysed on ice in GFP immunoprecipitation buffer (50 mM Tris-HCl, pH 7.6, 150 mM NaCl, 1% NP-40, 0.5% deoxycholate, 100 μl/ml HALT™ Protease/Phosphatase inhibitor cocktail EDTA free, 1 mM PMSF). GFP was immunoprecipitated with GFP-Trap_A (ChromoTek); 1/3 of the lysates were kept for WCL (whole cell lysates) analysis and quantification of total protein levels. Immunoprecipitates were washed four times in GFP-immunoprecipitation buffer and fractionated by SDS-PAGE, following the protocol in ref. 14. Briefly, 30 μg were separated via SDS-page using 4–15% Mini-PROTEAN® TGX™ Precast Protein Gels (Bio-Rad) and transferred to Immobilon-P PVDF (poly-vinylidene fluoride) membranes. PVDF membranes were blocked with 5% nonfat dry milk (wt/vol) in TBS-T buffer (0.1% vol/vol of Tween-20 in TBS buffer solution, #1706435 Bio-Rad) for 1 h at room temperature, and subsequently incubated for 1 h at room temperature or overnight at 4 °C with antibodies in 3% nonfat dry milk (wt/vol) in TBS-T buffer. Antibodies used were: GFP (Abcam # ab290) 1:2000; β-catenin (BD Transduction Laboratories #610154) 1:2000. After primary antibody incubation, PVDF membranes were washed in TBS-T, 2.5% nonfat dry milk (wt/vol) in TBS-T buffer, and 5% nonfat dry milk (wt/vol) in TBS-T buffer (10 min each). After these three washings, the membranes were incubated with the appropriate horseradish peroxidase-conjugated antibodies in 3% nonfat dry milk (wt/vol) in TBS-T buffer at a dilution of 1:5000. After secondary antibodies incubation, membranes were washed in TBS-T (3 × 10 min) and then incubated for 3–5 min with SuperSignal™ West Pico Chemiluminescent Substrate. Membranes were imaged with the Chemidoc MP Gel Documentation System (Bio-Rad). In case of re-probing by different antibodies, PVDF membranes were stripped using Restore Western Blot Stripping Buffer (21059, Thermo Scientific) for 15 min, washed in TBS-T for 10 min, and blocked in 5% non-fat dry milk (wt/vol) in TBS-T buffer at 4 °C overnight or for 1 h at RT.

## Single-molecule detector construct generation

Detailed sequence of the single-molecule detector construct for single-molecule manipulation experiments is shown below corresponding to Fig. 4A in the main text. The N-terminus and C-terminus of the single-molecule detector feature a biotinylated avi-tag and spy-tag, respectively. This configuration enables the specific tethering of the detector between a streptavidin-DNA-coated superparamagnetic bead and a spy-catcher-coated coverslip surface (see Fig. 4a). Acting as a spacer, a DNA handle (572 bp) coats the bead, maintaining distance between the bead and the coverslip surface. Force is applied to the detector via magnetic tweezers[85,86]. There is a short flexible linker (GGGSG) between two neighboring components. The plasmid is expressed in

Escherichia coli BL21 (DE3) cultured in LB-media with D-Biotin (Sigma-Aldrich), and affinity purified through his-tag (Supplementary Fig. 13a).

avi-$\beta$-cat-VBS-FH1-VD1-I27-I27-spy sequence

GLNDIFEAQKIEWHEGGGSGMATQADLMELDMAMEPDRKAAVSH
WQQQSYGGGSGEFPPAPPLPGDSGTIIPPPPPAPGDSTTPPPPPPPPPPPPPPL
PGGVCISSPPSLPGGTAISPPPPLSGDATIPPPPPLPEGVGIPSPSSLPGGTAIP
PPPPLPGSARIPPPPPPPLPGSAGIPPPPPPLPGEAGMPPPPPPLPGGPGIPPPP
PFPGGPGIPPPPPGMGMPPPPPFGFGVPAAPVLPGSGGGSGPVFHTRTIESI
LEPVAQQISHLVIMHEEGEVDGKAIPDLTAPVAAVQAAVSNLVRVGKETVQ
TTEDQILKRDMPPAFIKVENACTKLVQAAQMLQSDPYSVPARDYLIDGSR
GILSGTSDLLLTFDEAEVRKIIRVCKGILEYLTVAEVVETMEDLVTYTKNLGP
GMTKMAKMIDERQQELTHQEHRVMLVNSMNTVKELLPVLISAMKIFVTT
KNSKNQGIEEALKNRNFTVEKMSAEINEIIRVLQLTSWDEDAWLEGGGSGL
IEVEKPLYGVEVFVGETAHFEIELSEPDVHGQWKLKGQPLAASPDAEIIEDG
KKHILILHNAQLGMTGEVSFQAANTKSAANLKVKELGGGSGLIEVEKPLYG
VEVFVGETAHFEIELSEPDVHGQWKLKGQPLAASPDAEIIEDGKKHILILHN
AQLGMTGEVSFQAANTKSAANLKVKELGGGSGAHIVMVDAYKPTK

## Single molecule manipulation and analysis

A vertical magnetic tweezers setup[85,86] was used for conducting in vitro protein stretching experiments. In magnetic tweezers experiments, the height of the molecule-tethered superparamagnetic bead from the coverslip surface was recorded. During linear force-increase/force-decrease scans with a loading rate of 1 pN/s and −0.1 pN/s used in our study, the stepwise bead height change was the same as the stepwise extension change of the molecule. It was because the force change over the time window (≤0.01 s, the temporal resolution of our setup) of the stepwise transition was negligible (≤0.001–0.05 pN). All experiments were performed in solution containing 1× phosphate-buffered saline, 1% bovine serum albumin, 2 mM dithiothreitol, and 10 mM sodium l-ascorbate at 37° ± 2°C.

At sufficiently low forces, it is expected that VD1 may fold through the 182 amino acid FH1 polypeptide linker to bind to β-catenin VBS, forming a β-catenin VBS−VD1 complex (looped state) (see Fig. 4a). If a β-catenin VBS-VD1 complex can resist forces above a few piconewtons (pN), complex dissociation will result in a significant stepwise extension change (equivalent to the bead height change, ΔH) that can be observed in experiments. However, a stepwise extension change could arise from several potential scenarios from unfolding of the two subunits of VD1 (VD1A and VD1B) to dissociation of the complex. When β-catenin VBS is not associated with VD1 (Supplementary Fig. 13d), the two VD1 subunits are under force, which undergo non-cooperative unfolding[44]. In such instances, the two unfolding steps are associated with step sizes of ~35 nm across the force range of 5 pN to 20 pN according to our previous studies[44]. Therefore, a stepwise extension change with a step size significantly greater than 35 nm over the force range can be attributed to dissociation of a β-catenin VBS−VD1 complex.

## Laser ablation of cell-cell junctions

For the measurements of cell-cell junction recoil upon laser ablation, MDCK cells were cultured on fibronectin-coated coverglass prepared as described above. Cells were transfected with the expression vectors for ZO-1 fused with mEmerald as marker for cell-cell adhesions, in conjunction with vinculin constructs as appropriate.

UV laser nanoscissor ablation of cell-cell junctions in MDCK epithelial monolayer was performed on a Nikon A1R MP laser scanning confocal microscopy, equipped with an ultraviolet laser (PowerChip PNV-0150-100, Team Photonics: 355 nm, 300 ps pulse duration, 1 kHz repetition rate)[14]. The beam for imaging was combined with the UV laser beam onto the optical axis of the microscope, through a customized optical path and a dichroic filter to enable imaging and ablation simultaneously. The position of the beam was controlled by a mirror mounted on two linear actuators (TRA12CC, Newport), and corrected using an actuator controller (ESP301-3G, Newport). A mechanical shutter (VS25S2ZM0, Uniblitz) enabled the control of the exposure time.

A custom ImageJ plug-in was used to control both the actuators and the shutter and allowed us to draw a line with motorized mirror drives to do the line ablation (2–3 μm line across the membrane). By using a 15 nW laser power focused at the back focal plane of the objective, we could ablate precisely at the z-plane of cell-cell junction with an exposure time of 350 ms. Images were acquired with a pinhole size of 74 μm, every 2 s, starting from 6 s before ablation and for 30 s after it.

Image analysis of the recoil speed was conducted using Fiji by using the MTrackJ plugin[87]. This plugin allows the tracking of the two edges of the cut in subsequent frames and extracts the coordinates of the two points in time. The recoil speed (μm/s) was defined as the rate of change of the two edges. The initial recoil speeds were measured using the first 2 s after ablation.

## Fluorescence recovery after photobleaching (FRAP) measurements

We used FRAP to probe β-catenin dynamics in cell-cell junctions in MDCK wt, MDCK a-catenin KD, and T12 rescue. For the measurements of β-catenin mobility by FRAP, MDCK cells were cultured on fibronectin-coated coverglass prepared by incubating 10 μg/mL of bovine fibronectin (F1141, Sigma) in PBS for 1 h at 37 °C in humidified atmosphere. Cells were transfected with the expression vectors for β-catenin fused with eGFP as marker for cell-cell junctions, in conjunction with vinculin constructs with mApple or mcherry tag as appropriate. Imaging was performed using a Nikon Eclipse Ti-E inverted microscope (Nikon, Japan) equipped with a spinning disk confocal unit, CSU-W1 (Yokogawa, Japan), an iLas[2] illumination system, and a ProEM HS EMCCD camera (Princeton instruments, USA). The objective lens used was a CFI Plan Apo 60X NA 1.45 oil immersion (Nikon).

Ninety consecutive images were acquired every second. After the fourth acquisition, photobleaching in an ROI was carried out by scanning the 488 nm laser at 100% power for 2 ms. Acquired images were background subtracted and corrected for photobleaching by double normalization method, as follow:

$$I\,norm(t) = \frac{I\,ref_{pre}}{I\,ref(t) - I\,back(t)} \bullet \frac{Ifrap(t) - I\,back(t)}{Ifrap\_pre} \tag{1}$$

where I(t) $_{norm}$ is the normalized intensity; I (t) $_{frap}$ is the measured average intensity inside the bleached spot; I (t) $_{ref}$ is the measured average reference (an unbleached adhesion) intensity; I (t) $_{back}$ is the measured average background intensity outside the cell; subscript _pre means the averaging of intensity in the corresponding ROI before bleach moment after subtraction of background intensity. For the full-scale calibration, we used the following formula:

$$I\,norm1(t) = \frac{I\,norm(t) - I\,norm(t\,bleach)}{I\,norm_{pre} - I\,norm(t\,bleach)} \tag{2}$$

where $t_{bleach}$ is the bleach time.

## Wound-model migration assay

PDMS (Sylgard 184) blocks of $2 \times 1 \times 0.5$ mm (L × h × h) were prepared by cutting them out from PDMS slab made at 1:15 crosslinker to base ratio. Block were then coated with antifouling agent to prevent cell attachment. This was achieved by incubating them in 0.2% F-127 Pluronic acid (Sigma Aldrich) for 2 h. blocks were then washed with MiliQ water 3 times and oven-dried for at least 10 min. Meanwhile, cell culture grade 6-wells were briefly activated using a plasma cleaner, silanized with APTES for 1 h by vapor deposition, and then functionalized with fibronectin (Corning) diluted in Carbonate/Bicarbonate solution (PH 8.2) for 1 h to achieve a final coating of 3 μg/cm² as described in ref. 88. PDMS blocks were placed at the center of each well of a 6-well. Cells were seeded on each side of the PDMS block in DMEM with 10% FBS and 100 units/mL of Penicillin/ Streptomycin (Life

Technologies Carlsbad, CA) and maintained in a humified incubator at 37 °C with 5% $CO_2$ till cells reached full confluency, forming an epithelial monolayer around the PDMS block.

After careful removal of the PDMS block and replenishment of media with fresh DMEM pre-equilibrated for temperature and CO2, dynamics of migration of MDCK cells were visualized with 10X phase objective (Nikon) by phase contrast microscopy using BioStation CT imaging system (Nikon) and PC3 cells were visualized using Muvicyte imaging system (Perkin-Elmer), equipped with an LMPlantFL N (Olympus) 20 × 0.4 N.A objective. Images were acquired every 20 min for 48 h. Segmentation of the images for the tissue was performed using a self-developed MATLAB code based on pixel variance of image. Analysis of the migration experiments was conducted using MATLAB PIV (Particle Image Velocimetry). Magnitudes of speed below 10 μm h$^{-1}$ were filtered out to reduce motion noise. Angular analysis has been performed using a bin of 30° angles. Angular frequency distribution of the motion was calculated by dividing the number of counts per each angle bin by the total number of counts. Relative angular speed magnitude was calculated by multiplying the frequency by the mean speed measured in each angle bin (Fig. 4i, j).

### Statistics and reproducibility

Plotting and statistical analysis for superresolution z-position measurements (Welch's t-test and 1-way ANOVA) and laser ablation data (1-way ANOVA, followed by pairwise Tukey test) were performed using OriginPro software (Northampton, MA). For protein z-position, data are presented as median, and the mean, median, and standard deviation (std.dev.), and n in Supplementary Table 1. Statistical analysis is reported in Supplementary Table 2. For laser ablation experiments, the mean, standard deviation, s.e.m, and n are indicated in Supplementary Tables 3, 4. Differences were considered significant at $P < 0.05$ (as stated in each individual figure legend). For FRAP measurements (Fig. 4), curves are fitted to a single exponential function, $I(t) = p(1-e^{-kt})$. Half-time is $ln2/k$ (Supplementary Table 5). Statistical analysis (1-way ANOVA) of $p$, mobility fraction are indicated in Supplementary Table 6. For migration assay, rose plot diagrams have been used to show the circular distribution of directional data. Statistical analysis of wound-model migration assays (Fig. 4i-j) is indicated in Supplementary Table 7. Analysis was performed by fitting the angular distributions of the single experiments using a gaussian fitting ($Y = $ Amplitude × exp($-0.5 ×$ ((X-Mean)/SD)$^2$)) to obtain the major angular component for the distribution of single experiment. One-way ANOVA with Tukey correction for multiple comparisons was used to perform statistical analysis of the means of the major angular component for each condition. For MD simulations, statistical analysis and statistics for free energy ΔG measurements for all dimers and trimers are in indicated in Supplementary Data 1, 2 and Supplementary Table 9.

Western blots and immunofluorescence microscopy were consisting of at least three independent repeats. All representative microscopy images are presented with quantification of the entire data set. Detailed information on replication of experiments can be found indicated on the plots and in their corresponding legends. For determination of sample size, SD from initial trials was used to estimate the sample size based on confidence interval calculations at confidence levels of 95%.

### Reporting summary

Further information on research design is available in the Nature Portfolio Reporting Summary linked to this article.

## Data availability

The data that support the findings of this study are available from the corresponding authors upon request. Source data are provided with this paper.

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

## Acknowledgements

We gratefully acknowledge funding support from ANID (PIA ACT192015 and Fondecyt 1210872 C.B., A.R.; Fondequip EQM210101 A.R.; Fondecyt 1221030 A.F.; Fondecyt 1211060 N.P.B.), the Puente 2022-13 from Pontificia Universidad Catolica de Chile and the Singapore Ministry of Education (AcRF Tier 2, MOE2015-T2-1-045 and MOE2016-T2-1-124 to P.K.; MOE2015-T2-1-116 and T2EP30220- 0033 to Y.T.; AcRF Tier 3, MOE-T3-2020-0001 to P.K.; MOE-T2EP50123-0008 to J.Y.), the Singapore National Research Foundation Competitive Research Program (NRF2012NRF-CRP001-084 to P.K.) and Mid-Sized Grant (NRF-MSG-2023-0001 to J.Y.), Ministry of Education under the Research Centers of Excellence program through the Mechanobiology Institute at National University of Singapore (A-0003467-01-00 and A-0003467-00-00 to J.Y.) and Singapore National Research Foundation-Israel Science Foundation joint grant (NRF2019-NRF-ISF003-2952 to P.K.). We are grateful to administrative staff at Institute for Biological and Medical Engineering and Faculty of Biology, Unidad de Microscopia Avanzada (UMA) at Pontificia Universidad Catolica de Chile and, in particular, Fernanda Garate. We thank Paramasivam Kathirvel, Yi Herng Sim, and Julian Tze Kang Siew for assistance with CRISPR/Cas9, Aransa Griñen Muñoz and Cesar Ramirez Sarmiento for technical support with the molecular dynamics, Daniela Salas and Daniel Navarro for laboratory management and Diego Pitta de Araujo (Mechanobiology Institute at National University of Singapore) for the beautiful artistic production of the schematic model in Supplementary Fig. 4a.

## Author contributions

N.M.C. performed the molecular modeling and conducted the associated image analysis. M.J.R. conducted biochemistry experiments. C.B., A.R., and Y.W. performed the superresolution imaging experiments and conducted data analysis. P.K. supervised the iPALM imaging experiments. C.B. and A.R. performed and analyzed FRET experiments. C.S.L.L. and K.F.N performed CRISPR/Cas9 genome editing and characterization. Y.M. and R.N. performed experiments on knock-out cell lines. C.B., Y.T., H.T.O. designed and C.B. and A.R. performed and analyzed laser ablation experiments. P.C.S. performed, and J.J.A., H.T.O., and A.R. analyzed migration experiments. X.L. designed single-molecule construct and purified the protein. J.L. performed single-molecule stretching experiments and analyzed data. J.Y. supervised the single-molecule experiments. B.T.G., N.P.B., and A.F. participated in the discussion and development on the molecular modeling methods and data analysis. A.R., C.B., and P.K. provided new reagents, analytical tools, and in-depth discussion. C.B. and A.R. designed and supervised the study and wrote the manuscript. All authors discussed the results and commented on the manuscript.

## Competing interests

The authors declare no competing interests.

## Additional information

[1]Laboratory for Molecular Mechanics of Cell Adhesion, Faculty of Biological Sciences, Pontificia Universidad Católica De Chile, Santiago, Chile. [2]Department of Physics, National University of Singapore, 117542 Singapore, Singapore. [3]Faculty of Biological Sciences, Pontificia Universidad Católica de Chile, Santiago, Chile. [4]Institute for Biological and Medical Engineering, Schools of Engineering, Medicine and Biological Sciences, Pontificia Universidad Católica de Chile, Santiago, Chile. [5]Faculty of Medical Sciences, Universidad de Santiago de Chile, Santiago, Chile. [6]Millennium Institute for Foundational Research on Data (IMFD), Santiago, Chile. [7]Mechanobiology Institute, Singapore, National University of Singapore, 117411 Singapore, Singapore. [8]Department of Organic Chemistry, School of Chemistry, Faculty of Chemistry and Pharmacy, Pontificia Universidad Católica de Chile, Santiago, Chile. [9]Department of Biological Sciences, National University of Singapore, Singapore 117543, Singapore. [10]School of Biosciences, University of Kent, Kent, Canterbury CT2 7NJ, UK. [11]Department of Biochemistry, Cell & Systems Biology, Institute of Systems, Molecular & Integrative Biology, University of Liverpool, Crown Street, Liverpool L69 7ZB, UK. [12]Department of Biomedical Engineering, College of Design and Engineering, National University of Singapore, 117543 Singapore, Singapore. [13]Graduate School of Engineering Science, Osaka University, Osaka, Japan. ✉e-mail: and.ravasio@gmail.com; cristina.bertocchi@gmail.com

