## [Peer Review File · Nature Communications]

Alternative molecular mechanics of force transmission at adherens junction via β -catenin-vinculin interaction.Editorial Note: This manuscript has been previously reviewed at another journal that is not operating a transparent peer review scheme. This document only contains reviewer comments and rebuttal letters for versions considered at *Nature Communications*.

Reviewer #1 (Remarks to the Author):

Multi-step conformational arrangements of vinculin/catenin complexes modulate force transduction at the adherens junction.

Summary:

This study investigates molecular mechanisms of force transmission at adherens junctions (AJs) between epithelial cells. The authors present an alternative model to the current view of α -catenin as the primary force-bearing linkage between the cadherin-catenin complex and F-actin cytoskeleton. Using super-resolution microscopy, molecular modeling, and functional assays, they provide evidence for direct vinculin- β -catenin interactions that can also bear mechanical force at AJs.

Previous comment and their response in revised version:

1. Reviewer Concern: In general, the Methods section heavily relies on referencing publications while relevant details associated with the methodology is lacking.

Author's Response: Additional experimental data and computational approaches (including a procedure for conducting Molecular Mechanics Poisson-Boltzmann Surface Area (MM-PBSA) calculations) have been added, as recommended.

Reviewer Comment: The updated Methods section now provides a much clearer and comprehensive description of the procedures used.

2. Reviewer Concern: Sufficient details on the AlphaFold-generated full-length vinculin model (Uniprot ID/AlphaFoldID) are lacking. For example, it is unclear how low-complexity regions in the structure are modeled. The AlphaFold predicted model of full-length vinculin (P18206) has regions modeled with low and very-low confidence score (such as the linker region between head and tail domains, 838-960), and it is unclear whether flexibility/dynamics of this linker region dictate the organization of head- and tail domains upon engagement to other binding partners. The same argument holds for the catenin AlphaFold model. The Vinculin Binding Domain (VBS) has low-confidence score on the AlphaFold data base. Comparison of the AlphaFold models using alternative approaches (e.g. homology based modeling) should be considered with some discussion regarding limitations of the AlphaFold generated models.

Author's Response: AlphaFold was used as a graphical representation tool to illustrate the position of domains, rather than flexible regions where AlphaFold may not provide an accurate depiction. Alternative modeling approaches, such as homology-based modeling, have now been included to generate more reliable structural information.

Reviewer Comment: The author's response addresses this concern, and the updated methods section now clearly outlines the sources of structural information and modeling tools used for different regions of the proteins. This clarification helps to mitigate any potential confusion and provides a more comprehensive description of the methodology.

3. Reviewer Concern: The α -catenin interaction sites on vinculin identified using computational approaches (S1, S2 and S3) should be compared with known binding sites. Experimental validation (mutagenesis followed by experimental verification) is needed to support newly proposed interactions.

Author's Response: New MD simulations and energy calculations were performed to provide more details for the proposed binding sites (S1, S2 & S3) and residues between vinculin and α -catenin.

Reviewer Comment: The new binding interfaces and energies predicted from computational approaches alone are not definitive without corroborating biochemical data.

Although the authors have expanded computational analyses of vinculin- α -catenin interactions in the revised manuscript, concerns still remain about the lack of experimental validation to support the predictions from modeling. Authors acknowledge the need for experimental validation of the computational results but defer this to future studies. However, some mutational analysis or binding assays are necessary to back up the modeling, especially for newly proposed interactions. Without experimental confirmation, the predictions that α -catenin can still bind the A50I vinculin mutant, or interact with the S2 site, remain speculative. Hence, concerns raised previously have not been adequately addressed here.

4. Reviewer Concern: The authors estimate protein-protein complex stability using NAMD energy profiles rather than the MMPBSA. The latter is better suited for calculating binding affinities and free energy changes in a solvent environment. NAMD calculates the energies of molecular systems by solving the classical equations of motion, whereas MMPBSA (Molecular Mechanics Poisson Boltzmann Surface Area) calculates the free energy of a complex of proteins and/or ligands using molecular mechanics and continuum electrostatics. Further, MMPBSA explicitly models the solvent environment, which can be important for understanding the effect of solvent on the energetics of the system. Additionally, NAMD energy values listed are quite confusing. In some places the energy is written as a positive entity (Fig. 6), though it is negative in the graphs.

Author's Response: MM-PBSA calculations were performed to estimate binding affinity between vinculin/ α -catenin, vinculin/ β -catenin, vinculin(A50I)/ α -catenin, vinculin(A50I)/ β -catenin, vinculin/ α -catenin(L344P), vinculin/ β -catenin(M8P) and vinculin/ α -catenin/ β -catenin. Additionally, the authors calculated the energy contribution of residues responsible for the main interactions.

Reviewer Comment: The authors carried out Molecular Mechanics Poisson-Boltzmann Surface Area (MMPBSA) calculations to estimate the binding affinities between various protein complexes. This change enhances the depth and reliability of the study, as MMPBSA is indeed more suitable for calculating binding affinities and free energy changes in a solvent environment.

4. Review Concern: The X-structure of α -catenin VBS-Vh1 (PDB: 4EHP) suggests that Vh1- α -catenin interactions create a large hydrophobic interface that buries nearly 25% of the VBS solvent-accessible surface area (over 2,300 Å²). These interactions include 12 hydrogen bonds and novel hydrophobic interactions (Rangarajan et al., J. Biol. Chem., 2012). However, the authors state that the association of catenin and Vh is primarily through coulombic interactions. This should be validated through in silico mutagenesis/functional studies.

Author's Response: The presence of hydrophobic interactions and hydrogen bond interactions associated with the X-ray structure of α -catenin VBS-Vh1 are now discussed in more detail with emphasis on novel coulombic interactions that have not been previously described.

Reviewer Comment: While the authors have now emphasized the importance of new coulombic interactions and identified key residues involved through MD studies, in silico mutagenesis studies should be performed to validate the contribution of these residues to the stability of the vinculin- α -catenin complex. Furthermore, functional validation studies, such as site-directed mutagenesis/binding analyses are needed to provide experimental evidence to support the computational findings.

5. Reviewer Concern: The investigators heavily rely on z-positioning measurements at AJs to support the hypothesis that β -catenin engages vinculin in the absence of α -catenin. This analysis requires formation of a ROI around the fluorophore where the median position is recorded. Is the difference in z-position affected by the signal intensity? For example, if there is an increase or decrease in vinculin or β -catenin at the membrane, proximal to the cadherin-catenin layer, could it affect the z-positioning readout? This is interesting considering differences in β -catenin levels at the membrane in the different cell lines (Fig. 4D-E).

Author's Response: Detailed SAIM annotation has now been included in the method section and additional image clarity and explanation provided.

Reviewer Comment: The revisions adequately tackle the technical points raised regarding SAIM data analysis. The clarifications on SAIM principles and added controls support the z-positioning measurements for vinculin- β -catenin binding. The added description in the Methods provides helpful clarification on how SAIM works and provide a satisfactory explanation about SAIM z-position measurements. The authors cite their recent book chapter on SAIM theory to justify how the z-position is calculated from the shape of the measured intensity curve versus incident angle, alleviating concerns about z-position baseline measurements. Image clarification and western blots showing expression levels address the query regarding potential effects of varying protein amounts across cell lines. The authors have effectively responded to the concerns and strengthened the study.

6. Reviewer Concern: As vinculin showed the largest z displacement, what are the vinculin expression levels in these cell lines? This is relevant especially in the context of Figure 3A. If the A50I mutation is used to inhibit localization to focal adhesions, then is an increase in total localization of vinculin at AJs is expected? What percent of total vinculin is going to the AJ? A western blot and corresponding RT-PCR experiment to show differences in vinculin expression in the supplemental materials would be beneficial.

Author's Response: Co-immunoprecipitation data have now been included for the different vinculin mutants to show the expression levels (Sup. Fig 9) and vinculin expression at cell-cell adhesions analyzed via image analysis (Sup Fig 5).

Reviewer Comment: Although differences in exogenous vinculin expression levels are shown (Sup Fig 4C), the author's demonstrate vinculin localization to cell-cell adhesions via co-IP (Sup Fig 9) and image analysis (Sup Fig 5). It has been demonstrated that the T12 A50I vinculin mutant does not strongly associate with cell-cell adhesions in the presence of α -catenin KD as seen in the main text. It would be helpful to determine the percentage of vinculin localizing to cell-cell adhesions with respect to total vinculin localization.

7. Reviewer Concern: The authors' hypothesis that β -catenin engages vinculin in its activated state following α -catenin depletion is based on findings in Fig 2A-C. As noted in previous studies, other vinculin interacting proteins (e.g. Ena/VASP) form distinct and independent mechanical connection with the actin cytoskeleton (Scott, J.A et al. (2005). *MBoC*, 17 3, 1085-95., Oldenburg, J. et. al. *Sci Rep* 5, 17225 (2015). <https://doi.org/10.1038/srep17225>). Have the authors performed any control experiments in Fig 2A-C to check the involvement/ role of these proteins alongside β -catenin?

Author's Response: Additional references and discussion are now included regarding the potential involvement of other mechanosensitive linkages in the main text.

Reviewer Comment: The authors have now expanded the discussion of key proteins including α -actinin, myosin VI, eplin, and afadin as possible alternative mechanosensitive linkages at adherens junctions in the revised manuscript, which now provides more context and balanced view of adherens junction mechanobiology.

8. Reviewer Concern:
Minor comments on computational modeling, MD simulations, and sequence analysis.

Reviewer Comment: The authors have adequately addressed minor concerns related to computational modeling and sequence analysis. Their revised manuscript now provides clearer explanations and additional details in response to the issues raised. Specifically, the authors have provided a more comprehensive description of description of MD simulations setup, sequence alignment of VBS1, VBS2, and VBS3, phylogenetic analysis, %similarity and %identity between the VBS of *catn1* and α -catenin and β -catenin, stability of selected salt-bridge interactions at the α -catenin – vinculin interface in MD trajectories, and stability of the D1/-VBS during 1 μ s MD simulations. Overall, the revisions have significantly strengthened the manuscript.

A. Summary of the key results:
Keys points are added above.

B. Originality and significance: if not novel, please include reference

The role/involvement of β -catenin in mechanical connection with the actin cytoskeleton and force transmission is not well understood (Peng, X., Cuff, L. E., Lawton, C. D. & DeMali, K. A. Vinculin regulates cell-surface E-cadherin expression by binding to beta-catenin. J. Cell Sci.

Authors in the current study propose a novel role/involvement of β -catenin in force transmission at cadherin based junctions.

C. Data & methodology: validity of approach, quality of data, quality of presentation

Standard number of experimental replicates (N, n) are now reported. Key details and explanation or method references are now added/expanded. In addition, the authors have streamlined the main figures and provided supportive Supplemental data in the revised submission.

D. Appropriate use of statistics and treatment of uncertainties

Standard statistical analysis that were missing in the previous version are now included. Authors have now performed standard statistical analysis test to validate Fig 1H-I, Fig 2A&C and Fig 4.

E. Conclusions: robustness, validity, reliability

The authors hypothesize that β -catenin forms a direct mechanical connection with the actin cytoskeleton via a direct interaction with vinculin. The supporting MD and cellular studies have been strengthened. However, central issues remain about lack of biochemical support for this hypothesis. Justification to explore/negate the involvement of known players such as Ena/VASP, zyxin and TES etc has now been included in the revised manuscript.

Proposed interaction sites on vinculin for interactions with α -catenin and β -catenin are based on modeling and MD simulation studies. Experimental support is still missing and needed for validation.

F. Suggested improvements: experiments, data for possible revision

See comments

G. References: appropriate credit to previous work?

Most studies are now included.

H. Clarity and context: lucidity of abstract/summary, appropriateness of abstract, introduction and conclusions.

The abstract concisely summarizes the main findings. The introduction explains the background and goals of the study. The conclusions discuss the impact of the results without over extending interpretation of their findings. Overall, these sections are written clearly and provide suitable framing for understanding the study and its significance.

Reviewer #3 (Remarks to the Author):

I am a new reviewer to this revised manuscript by Morales-Camilo. The manuscript focuses on how vinculin, in the absence of alpha-catenin can bind to beta-catenin. The manuscript uses a variety of computational analysis of protein structure, as well as experimental models to increase understanding of how vinculin interacts with catenins at cell-cell adhesions.

Major concerns:

1. In the last response to reviewers the authors mentioned that there were no alpha-catenin KO MDCK cell lines available.

This is not correct. MDCK alpha-catenin KO cells using crispr were developed by Ozawa, Biology Open 2018. A more recent publication (Bejar-Padilla, Molecular Biology of the Cell, 2022) developed a similar line. Although the authors show clear differences between WT and KD cell lines, it is not clear if the residual alpha-catenin in the KD cells is impacting things. It would have been much stronger to demonstrate the vinculin to beta-catenin interaction in alpha-catenin KO cells, to demonstrate alpha-catenin is not mediating this interaction.

2. In this revision the authors have added new details about PC3 cancer cells (prostate) suggesting that these cells may be collectively migrating using the beta-catenin vinculin interaction, given low alpha-catenin levels. To me this seems speculative and no efforts were made to modulate these cells to test this hypothesis.

3. Does vinculin T12 localize more strongly to cell-cell adhesions? I think this could be an important point to assess, since vinculin must traffic between focal adhesions and cell-cell adhesions, it needs to be made clear if the T12 vinculin to beta-catenin interaction is a result of the T12 mutations directly, or instead indirectly a result of the vinculin being more present at cell-cell adhesions.

4. Figure 4G: I had a hard time understanding if the differences in migration were significant. Are the migration velocities different? I watched the movies, but was unsure. Also the graphs shown, with regard to direction have no error bars and it is difficult to know if the differences are meaningful.

5. The data presented in Figure 4A (recoil speed) is a different finding than a recent paper (Mezher, Biophys J 2023) that showed no major differences in inter-cellular forces in alpha-catenin KO cells. While these are two different methods (and may be measuring different things) the authors should directly discuss this other manuscript. I note the manuscript is already cited (#35) in the introduction. The Mezher manuscript also cites the pre-print of the present paper mentioning that "only a minor decrease in cell-cell tension as assessed by the retraction of the ablated ends of cell-cell contacts". I am not sure if the authors agree or disagree with the Mezher paper calling the effect "minor"--but I think they should discuss it nonetheless.

General Responses to Reviewers' Comments II

We thank the reviewers for their highly positive comments and for dedicating their time to carefully review our work. We also are grateful to the reviewers for recognizing that our work is impactful, and it will be well received by the large cell biology and mechanobiology communities.

In the second revision of the manuscript, along with extensive experimental and conceptual work to address the reviewers' comments, we performed magnetic tweezer experiments to provide a direct evidence of a force-bearing functional interaction between β -catenin and vinculin in absence of α -catenin. We believe this provides the final proof that validates the main message of our work. These results have been included in the new manuscript as Figure 4A-B and Supplementary Figure 13 and they have been included in the relevant part of the results and discussion that now read:

Lines 336-359 - " To investigate this, we performed in vitro single-molecule magnetic tweezer experiments using a single-molecule detector containing β -catenin VBS and vinculin VD1 separated by formin FH1 domain that functions as unstructured spacer (Figure 4A, top and Supplementary Figure 13A). At sufficiently low forces, it is expected that, if β -catenin VBS can effectively interact with VD1, a VBS—VD1 complex is formed (Figure 4A, bottom). Such interaction should also be able to resist forces above a few piconewtons (pN) before dissociating resulting in a significant stepwise extension change (equivalent to the bead height change, ΔH) that can be observed in experiments (Figure 4B). We conducted experiments using 26 independent tethers and observed dissociation of the β -catenin VBS—VD1 complex in eight cases (17 dissociation events from 8 individual tethers, exemplary experiments in Figure 4B, inset and Supplementary Figure 13). This reflects a relatively low ($\sim 30\%$) probability of complex formation, which reconciles with the calculated free energy from MM-PBSA analysis for vinculin/ β -catenin complex (-28.9 Kcal/mol) as compared to that of the vinculin/ α -catenin complex (-63.2 Kcal/mol). Within the reported dissociation events, we could observe a large variability in ΔH in response to forces ranging from 7 to 16 pN at a loading rate of 1 pN s⁻¹ as indicated by the counts of single events as a function of the applied force and step size (Figure 4B, top and right histograms). Assuming a similar bending persistence of ~ 0.8 nm for the polypeptide polymer from VD1 unfolding and FH1 unlooping, we can predict the force-dependent step sizes resulting from VBS-VD1 dissociation and from VD1 subunit unfolding. Based on this calculation, three scenarios are possible: dissociation of VBS-VD1 dissociation without concurrent VD1 subunit unfolding (unlooping, theoretical $\Delta H \approx 40$ -50 nm), dissociation with concurrent unfolding of one VD1 subunit (unlooping and partial unfolding, theoretical $\Delta H \approx 60$ -90 nm), and finally dissociation with concurrent complete unfolding of VD1 (unlooping and unfolding, theoretical $\Delta H > 100$ nm). In conclusion, these findings indicate that, once established, β -catenin VBS—VD1 complex can withstand physiologically-relevant forces (ranging from 7 to 16 pN), thereby facilitating a mechanically stable force transmission between β -catenin and the cytoskeleton network through association with vinculin."

Lines 465-473 - " Our results clearly demonstrate that, in absence of α -catenin, a functional direct interaction between β -catenin and non-autoinhibited vinculin can form to sustain physiologically relevant forces (Figure 4B). A loading rate in the order of pN s⁻¹ is anticipated by force-transmission to membrane receptors that orientates the corresponding force-transmission supramolecular linkage (as discussed in63) and mechanical unfolding of force-bearing domains (such as vinculin domains64) due to actomyosin contraction. The observed forces of 7 to 16 pN for the disruption of the β -catenin VBS-VD1 complex are in the similar range where mechanical unfolding of vinculin domains occurs64. Therefore, this complex, once formed, is able to support force transmission that could lead to a downstream cascade of mechanotransduction events mediated by vinculin."

We also clarified what is the main and most novel discovery of our manuscript by changing the title of the manuscript which now reads:

A novel role for β -catenin in force transmission at cadherin-based adhesions.

Similarly, we changed the title of the subchapters in the results session to better specify this point and avoid overstatements.

Additionally, we also generated an MDCK α -catenin KO cell line to verify consistency of recruitment of vinculin at the AJ in complete absence of α -catenin (see Supplementary Figure 6).

We believe this version conclusively validates the message of our manuscript and ultimately addresses all remaining open questions. We thank the reviewers for their positive and constructive comments which encouraged us to further corroborate our findings.

Below we provide a point-by-point reply to the reviewers' comments. We highlighted in red the reviewers' comments, in black our response and in blue the changes made in the manuscript, figures and supplementary materials.

Specific Responses to Reviewer's comment:

Reviewer #1

We are glad to see that reviewer 1 is pleased with the large majority of our responses and that they agree that the extensive work done during our first revision has greatly improved the manuscript. Reviewer 1 further requested A) experimental validation through site mutagenesis of the results of vinculin/ α -catenin molecular modeling and B) the quantification of vinculin specific localization at the cell-cell junction.

Point A

Previous discussions in red:

3. Reviewer Concern: The α -catenin interaction sites on vinculin identified using computational approaches (S1, S2 and S3) should be compared with known binding sites. Experimental validation (mutagenesis followed by experimental verification) is needed to support newly proposed interactions.

Author's Response: New MD simulations and energy calculations were performed to provide more details for the proposed binding sites (S1, S2 & S3) and residues between vinculin and α -catenin.

Reviewer Comment: The new binding interfaces and energies predicted from computational approaches alone are not definitive without corroborating biochemical data.

Although the authors have expanded computational analyses of vinculin- α -catenin interactions in the revised manuscript, concerns still remain about the lack of experimental validation to support the predictions from modeling. Authors acknowledge the need for experimental validation of the computational results but defer this to future studies. However, some mutational analysis or binding assays are necessary to back up the modeling, especially for newly proposed interactions. Without experimental confirmation, the predictions that α -catenin can still bind the A50I vinculin mutant, or interact with the S2 site, remain speculative. Hence, concerns raised previously have not been adequately addressed here.

4. Review Concern: The X-structure of α -catenin VBS-Vh1 (PDB: 4EHP) suggests that Vh1- α -catenin interactions create a large hydrophobic interface that buries nearly 25% of the VBS solvent-

accessible surface area (over 2,300 Å²). These interactions include 12 hydrogen bonds and novel hydrophobic interactions (Rangarajan et al., J. Biol. Chem., 2012). However, the authors state that the association of catenin and Vh is primarily through coulombic interactions. This should be validated through in silico mutagenesis/functional studies.

Author's Response: The presence of hydrophobic interactions and hydrogen bond interactions associated with the X-ray structure of α -catenin VBS-Vh1 are now discussed in more detail with emphasis on novel coulombic interactions that have not been previously described.

Reviewer Comment: While the authors have now emphasized the importance of new coulombic interactions and identified key residues involved through MD studies, in silico mutagenesis studies should be performed to validate the contribution of these residues to the stability of the vinculin- α -catenin complex. Furthermore, functional validation studies, such as site-directed mutagenesis/binding analyses are needed to provide experimental evidence to support the computational findings.

Our response, second revision:

We entirely agree with the reviewer that in silico and in vitro site mutagenesis and biochemical confirmation would further validate the MD analysis to confirm the predicted sites of interactions for the vinculin/ α -catenin dimer. In the present version, we discussed the experimental validation of the interaction of α -catenin VBS with S1 (VD1a) and S2 (VD1b) sites, which were previously published by co-authors of the current manuscript (Le et al. Sci Adv. 2019). While the cited work does not elucidate the exact binding sites, it supports the general molecular arrangement that our MD analysis hints to.

Furthermore, we would kindly point out that the site mutagenesis of each of the residues predicted by MD would require a huge investment of time and resources (also considering that the work is conducted in a resource-limited environment) to validate a point that is only partially relevant for the present work. As indicated by the new title, which as a matter of fact was inspired by the reviewer's comment regarding the novelty of our work, our manuscript focuses on the newly discovered vinculin/ β -catenin functional interactions. Importantly, the reviewer seems to be fully satisfied by the extensive body of evidence provided (super-resolution microscopy, FRAP, laser ablation, co-immunoprecipitation) and now this key point is further strengthened by additional single molecule magnetic tweezer experiments. We also wish to stress the fact that, in this work, we used MD analysis as a power comparative tool to make prediction of the relative importance (based on energetic considerations) and possible configurations that allowed to propose the functional model reported in figure 6.

Nevertheless, we fully agree with the reviewer that our in-silico data alone are not sufficient to make strong statements regarding the specific residues mediating α -catenin-vinculin interaction. Also, residues reported for the other interactions would require similar scrutiny and further experimental validation, which would account for a tremendous amount of work, which we can only defer to a future publication. Thus, we further emphasize throughout the manuscript that experimental evidence is needed to validate the MD analysis and we trust that reviewer will consider this as an agreeable compromise, also in view of the fact that the manuscript has been transferred to an interdisciplinary journal, where a large portion of the audience may not be as interested in the molecular details.

Major changes to the manuscript can be found at:

Abstract

Lines 42-47 - "...we demonstrated that when α -catenin is depleted, β -catenin can partially occupy the α -catenin-binding site on vinculin head and bear force by direct connection with vinculin in its open conformation. While, by molecular dynamics, we could predict that when α -catenin occupies the groove on vinculin head, β -catenin could bind to the site involved in vinculin head-to-tail interaction, preventing its closure and thus maintaining the ability of vinculin to transmit large forces."

Results

Lines 146-163 - "During 1 μ s of molecular dynamics (MD) simulation, α -catenin VBS maintained an interaction with S1 and S2 of vinculin (Figure 1E, Supplementary Video 1 and Supplementary Figure 2A). This agrees with previous single molecule magnetic tweezer studies⁴⁴ from authors of this paper that experimentally demonstrated the viability of the formation of a complex between α -catenin VBS and S1 (VD1a) and S2 (VD1b) of vinculin. While the experimental verification of the exact residues mediating this interaction is still missing, MD analysis for the stability of the complex predicts the involvement of intermolecular hydrogen bonds (S349, Y351, S323, E336 of α -catenin and Q19, R105, K170, N53 of vinculin), hydrophobic stabilization of conserved non-polar residues motif inside the groove of S1, and salt bridge interactions (E60, E66 of vinculin and R332, R326 of α -catenin) (Figure 1E). Furthermore, Adaptive Poisson-Boltzmann Solver (APBS) analysis of the electrostatic surface potential shows a high negative charged area (red) on vinculin between S1 and S2 and, a low electron density area (blue) on the central part of α -catenin VBS (Figure 1F). Based on the general interactions in the dimer vinculin/ α -catenin, we performed a per-residue Molecular Mechanics with Poisson-Boltzmann and Surface Area solvation (MM-PBSA) analysis to calculate the global free energy ($\Delta G_{\text{vinculin}/\alpha\text{-catenin}} = -63,2$ Kcal/mol) of the dimer (the residues that have the largest energetic contribution to the complex stabilization are represented in Figure 1G and the short mean distance along the MD in Supplementary Figure 3). These results are in line with previously published papers by W.Weis⁴² and our groups⁴⁴, and support the involvement of S1 (VD1a) and S2 (VD1b) in the interaction with α -catenin VBS."

Point B

Previous discussions in red:

6. Reviewer Concern: As vinculin showed the largest z displacement, what are the vinculin expression levels in these cell lines? This is relevant especially in the context of Figure 3A. If the A50I mutation is used to inhibit localization to focal adhesions, then is an increase in total localization of vinculin at AJs is expected? What percent of total vinculin is going to the AJ? A western blot and corresponding RT-PCR experiment to show differences in vinculin expression in the supplemental materials would be beneficial.

Author's Response: Co-immunoprecipitation data have now been included for the different vinculin mutants to show the expression levels (Sup. Fig 9) and vinculin expression at cell-cell adhesions analyzed via image analysis (Sup Fig 5).

Reviewer Comment: Although differences in exogenous vinculin expression levels are shown (Sup Fig 4C), the author's demonstrate vinculin localization to cell-cell adhesions via co-IP (Sup Fig 9) and image analysis (Sup Fig 5). It has been demonstrated that the T12 A50I vinculin mutant does not strongly associate with cell-cell adhesions in the presence of α -catenin KD as seen in the main text. It would be helpful to determine the percentage of vinculin localizing to cell-cell adhesions with respect to total vinculin localization.

Our response second revision:

As suggested by the reviewer we have analyzed relative distribution of vinculin at the cell-cell junction versus the total vinculin. The results of this analysis are shown in Supplementary Figure 5B and C and support the lower recruitment of vinculin T12-A50I mutant to cadherin-mediated adhesions respect to Vinculin wt and T12 mutant. This has been further validated using a newly generated α -catenin KO cell line. We thank the reviewer for this suggestion as the quantification allowed us to validate the results obtained by MD (Figure 3C-D), co-IP (Suppl. Figure 10), SAIM (Figure 3A) and the functional analysis shown in Figure 4C to J.

The main text reports this analysis at lines 215-217:

“This ^[vinculin localization] was verified in both α -catenin knockdown as well as knockout conditions confirming that the recruitment of active vinculin to the cadherin-catenin layer is not due to the minimal amount of leftover α -catenin present in the MDCK α -catenin KD (Supplementary Figure 5 and 6).”

And at lines 401-404:

“Importantly, equal amounts of vinculin wt and vinculin T12 in MDCK wt and MDCK α -catenin KD were measured at the AJ (Supplementary Figure 5B and C), thus proving that results from laser ablation, FRAP and wound-model assay should be interpreted as consequence of the type of vinculin configuration/mutation and not to the relative amount of protein at the AJ.”

Reviewer #3:

We are grateful to the reviewer for stepping in and providing us with their valuable feedback and expert comments. Much appreciated!

Major concerns:

1. In the last response to reviewers the authors mentioned that there were no alpha-catenin KO MDCK cell lines available.

This is not correct. MDCK alpha-catenin KO cells using crispr were developed by Ozawa, Biology Open 2018. A more recent publication (Bejar-Padilla, Molecular Biology of the Cell, 2022) developed a similar line. Although the authors show clear differences between WT and KD cell lines, it is not clear if the residual alpha-catenin in the KD cells is impacting things. It would have been much stronger to demonstrate the vinculin to beta-catenin interaction in alpha-catenin KO cells, to demonstrate alpha-catenin is not mediating this interaction.

Our response:

We thank the reviewer for pointing out the availability of MDCK α -catenin KO. We tried in the past, but failed and wrongly assumed that a complete α -catenin KO was not feasible. Thus, in collaboration with co-authors of this paper, we generated an MDCK α -catenin KO cell line using CRISPR/Cas9 technology (Supplementary Figure 6). While it would not be feasible to re-do all experiments using this new cell line, we used it to validate the important point whether vinculin can still be recruited at the AJ in complete absence of α -catenin. Supplementary figure 6 (MDCK α -catenin KO) and Supplementary figure 5 (α -catenin KD) demonstrate that vinculin recruitment does not require residual expression of α -catenin. This result, combined with the notion derived from the newly presented single molecule magnetic tweezer experiments that β -catenin directly and

functionally interacts with vinculin (Figure 4A and B), strengthens the conclusions derived from the functional studies shown in Figure 4C to J.

This is included in the new manuscript at lines 215-217 that now read:

“This ^[vinculin localization] was verified in both α -catenin knockdown as well as knockout conditions confirming that the recruitment of active vinculin to the cadherin-catenin layer is not due to the minimal amount of leftover α -catenin present in the MDCK α -catenin KD (Supplementary Figure 5 and 6).”

And at lines 401-404 that now read:

Importantly, equal amounts of vinculin wt and vinculin T12 in MDCK wt and MDCK α -catenin KD were measured at the AJ (Supplementary Figure 5B and C), thus proving that results from laser ablation, FRAP and wound-model assay should be interpreted as consequence of the type of vinculin configuration/mutation and not to the relative amount of protein at the AJ.

2. In this revision the authors have added new details about PC3 cancer cells (prostate) suggesting that these cells may be collectively migrating using the beta-catenin vinculin interaction, given low alpha-catenin levels. To me this seems speculative and no efforts were made to modulate these cells to test this hypothesis.

The reviewer is certainly correct to point out that this observation is speculative and demonstration of pathological relevance of our investigation would still requires further in-depth investigation. However, we wish to reassure the reviewer that we did not base our statement on a wild guess/assumption. A main effort of our laboratory is now directed to demonstrate the pathological relevance of β -catenin/vinculin in prostate cancer cell lines. Preliminary data show that, as expected, cell-cell junctions in PC3 are not as stable as those of normal epithelial cells (due to the absence of α -catenin). Consequently, a small degree of single-cell scattering along with cell streaming is still detectable. Interestingly, rescue experiments where α -catenin is knock-in in PC3 cells show a transition to strong collective migration and complete inhibition of residual single cell migration, as shown in the figure below.

As we agree with the reviewer, we have modified the manuscript to clearly indicate that this point remains speculative at this time and further analysis would be needed. However, we do not feel it

would be appropriate to entirely remove it from the manuscript as the supplementary video 11 was specifically requested and accepted by reviewer 1 who asked to suggest possible physiological or pathological scenarios relevant to our findings.

In principle, we could also agree to remove entirely this point from the manuscript (also because we are working on expanding this interesting point in future publications). Thus, we brought it to the attention of the editor who we believe should also arbitrate eventual divergence of opinion between the reviewers.

To avoid misleading the readers, we clarified as honestly as we could how to interpret those data.

The main text has been modified as follows at line 404-408:

“For instance, such novel function could potentially explain how PC3 cells, which are derived from prostate cancer, are capable of migrating collectively (Supplementary video 11) despite lacking α -catenin (expression levels in Supplementary Figure 4C-D). However, this observation remains only suggestive of possible conditions where force transmission by the β -catenin/vinculin complex can possibly explain yet unresolved pathological scenarios.”

3. Does vinculin T12 localize more strongly to cell-cell adhesions? I think this could be an important point to assess, since vinculin must traffic between focal adhesions and cell-cell adhesions, it needs to be made clear if the T12 vinculin to beta-catenin interaction is a result of the T12 mutations directly, or instead indirectly a result of the vinculin being more present at cell-cell adhesions.

We thank the reviewer for raising this important point. We quantified the relative distribution of exogenous vinculin at the AJ vs cytosolic levels (indicative of protein recruitment) at cell-cell junction in MDCK wt and MDCK α -catenin KD transfected with vinculin wt, T12 and T12-A50I. Only T12-A50I showed a lower level of recruitment at the AJ (Supplementary figure 5B). The results of vinculin T12 was also corroborated using MDCK α -catenin KO cells (Supplementary figure 6C, E, D). These results prove that the functional studies here reported are to be attributed to the type of vinculin configuration/mutation rather than its amount at the AJ.

This has been included in the main text at lines 401-404 that now read:

“Importantly, equal amounts of vinculin wt and vinculin T12 in MDCK wt and MDCK α -catenin KD were measured at the AJ (Supplementary Figure 5B), thus proving that results from laser ablation, FRAP and wound-model assay should be interpreted as consequence of the type of vinculin configuration/mutation and not to the relative amount of protein at the AJ.”

4. Figure 4G: I had a hard time understanding if the differences in migration were significant. Are the migration velocities different? I watched the movies, but was unsure. Also the graphs shown, with regard to direction have no error bars and it is difficult to know if the differences are meaningful.

We apologize for the poor representation of our wound-model assays. Following the reviewer's advice, we have largely modified this session. Most importantly, we have:

- 1) Included the standard deviation to the relevant figure panels (now Figure 4I and J);
- 2) Provided the relative angular speed (which describes the difference in tissue motion for each angular sector) in addition to the mere frequency distribution;
- 3) Included statistical analysis that demonstrates that MDCK wt and MDCK α -catenin KD with vinculin T12 are statistically very similar, whereas MDCK α -catenin KD behave less collectively;

- 4) Provided the quiver representations of the PIV analysis in the supplementary movies. We believe that this last quite clearly shows loss of collective migration (laminar flow) in MDCK α -catenin KD as compared to the other conditions, which look quite similar to each other.

These changes can be found various parts of the main text, figures and supporting material:

- Figure 4I and J now substitute Figure 4G;
- New Supplementary Movies containing the PIV quiver have been uploaded (Supplementary Video 8A, B and C);
- Supplementary Table 4 reports the statistical analysis;
- Experimental method, PIV analysis and statistics have been expanded in the methods.

The relevant paragraph in the main text at lines 380-401 now reads:

“Finally, to further test force transmission between cells under these conditions and evaluate their large-scale consequences, we probed for the ability of cells to maintain collective migration in a wound-model assay (Figure 4I-J, Supplementary table 4 and Supplementary Video 8A-C). Particle Image Velocimetry (PIV) analysis of these experiments, which vectorially represents the flow of cells, showed that expectedly MDCK wt cells migrate cohesively by coordinated collective migration (Supplementary Video 8A) and the preponderant frequency distribution (Figure 4I, left) as well as the relative angular speed (Figure 4J, left) being perpendicular to the migrating front (i.e., 0°). In contrast, chaotic motility and decreased coordination was observed in MDCK α -catenin KD as consequence of partial loss of force transmission between cells (Supplementary Video 8A) as indicated by the lower frequency distribution and relative angular speed perpendicular to the front of the tissue edge (i.e., 0°) (Figure 4I and J, middle panels). Interestingly, Supplementary Video 8C and right panels in Figure 4I and J demonstrate that expression of T12 in α -catenin KD cells was able to largely rescue the cohesive migration phenotype seen in wildtype cells. It must be noted that by experimental design, contact inhibited cells, such as MDCK cells, will be prevented from migrating toward the bulk of the tissue and thus a preferential motion perpendicular to edge of the tissue (along the x axis) is to be expected. As result, motion is directionally skewed, and statistical analysis has reduced degrees of freedom. Nonetheless, statistical analysis indicates high correspondence between mode of migration of MDCK wt and MDCK α -catenin KD transfected with vinculin T12 mutant (p values = 0.81 and 0.97 for the angular frequency and mean speed magnitude, respectively – Supplementary Table 4). Concomitantly, low similarity was observed when comparing MDCK α -catenin KD with the other samples ($0.17 < p \text{ values} < 0.37$ for all samples when compared to α -catenin KD).”

5. The data presented in Figure 4A (recoil speed) is a different finding than a recent paper (Mezher, Biophys J 2023) that showed no major differences in inter-cellular forces in alpha-catenin KO cells. While these are two different methods (and may be measuring different things) the authors should directly discuss this other manuscript. I note the manuscript is already cited (#35) in the introduction. The Mezher manuscript also cites the pre-print of the present paper mentioning that "only a minor decrease in cell-cell tension as assessed by the retraction of the ablated ends of cell-cell contacts". I am not sure if the authors agree or disagree with the Mezher paper calling the effect "minor"--but I think they should discuss it nonetheless.

In a nutshell, we agree with the authors that alternative paths bypassing α -catenin can mediate force transmission at the AJ. However, we disagree with their view that consequently α -catenin is not important.

It has to be noted that indeed laser ablation measures elastic tension built along the cell-cell junctions whereas the method described by Mezher et al., which is derived from the group of Margaret Gardel (Maruthamuthu et al. PNAS 2011), measures normal forces perpendicular to the AJ. Another main difference is that the forces we measure are in confluent epithelial tissues, whereas they perform their measurements in cell aggregates composed by 2-3 cells, where the junction is likely to be less mature as compared to full confluency.

Despite these differences, we do not understand what justified the authors assessment that our laser ablation experiments show "only minor" effects. In absence of α -catenin we see relevant and significant reduction in tension. Similarly wound model assays are also in agreement with this loss of function. Furthermore, several groups, including our, consistently reported this effect over the years and in several publications (e.g., Ravasio et al. Integrative Biol. 2015, Seddeki et al. MboC 2018, Barry KA et al J Cell Sci. 2014, etc).

At last, the same authors seem to have contradictory results as figure 5 of the same paper show that α -catenin is crucial to maintain cohesiveness of the tissue in biaxial stretching experiments.

We have explicitly argued about this point in the discussion of the paper that now at lines 451-456 reads:

"While this view [central importance of \$\alpha\$ -catenin in force transmission at the AJ] has been recently challenged³⁵, our results, based on direct measurement of tensional state of AJ, clearly demonstrate a significant loss of tension in α -catenin KD cells (Figure 4D-E), confirming the central role of α -catenin under physiological conditions. However, growing body of evidence suggests that alternative pathways could mediate force transmission at the AJ²⁹⁻³⁵. Here, we demonstrate that β -catenin can form a physical connection with open vinculin, capable of rescuing force transmission in absence of α -catenin."

Reviewer #3 (Remarks to the Author):

I think the authors have done an excellent job of addressing my and the other reviewer's comments.

The revised manuscript clearly demonstrates there is a vinculin-beta catenin interaction and that this interaction is capable of mechanical force transmission. This is impactful because alpha-catenin has been assumed to be the only linker for connecting vinculin to adherens junctions.

Reviewer #4 (Remarks to the Author):

In the revised submission, authors have addressed the concerns raised during the previous round of review. And the inclusion of single-molecule magnetic tweezer experiments in the revised version provides direct evidence of a force-bearing functional interaction between β -catenin and vinculin in the absence of α -catenin, further corroborating their findings. This data, along with the comprehensive evidence from super-resolution microscopy, FRAP, laser ablation, and co-immunoprecipitation, convincingly establishes the novel role of β -catenin in force transmission at cadherin-based adhesions.

Regarding the previous concern about vinculin expression levels in different cell lines and their potential impact on the observed z-displacement, the authors have adequately addressed this by analyzing the relative distribution of vinculin at the cell-cell junctions versus the total vinculin. By analyzing the relative distribution of vinculin at the cell-cell junctions versus the total vinculin, they have demonstrated that the lower recruitment of the vinculin T12-A50I mutant to cadherin-mediated adhesions compared to wild-type vinculin and the T12 mutant is not due to differences in expression levels.

While the lack of experimental validation of the molecular dynamics simulations for the specific residues mediating α -catenin-vinculin interactions remains a limitation, the authors have acknowledged this and emphasized the need for future experimental validation. Considering the extensive experimental evidence supporting the main conclusions of the study, I believe that this limitation does not detract from the overall impact of the work.

REVIEWERS' COMMENTS

Reviewer #3 (Remarks to the Author):

I think the authors have done an excellent job of addressing my and the other reviewer's comments.

The revised manuscript clearly demonstrates there is a vinculin-beta catenin interaction and that this interaction is capable of mechanical force transmission. This is impactful because alpha-catenin has been assumed to be the only linker for connecting vinculin to adherens junctions.

Reviewer #4 (Remarks to the Author):

In the revised submission, authors have addressed the concerns raised during the previous round of review. And the inclusion of single-molecule magnetic tweezer experiments in the revised version provides direct evidence of a force-bearing functional interaction between β -catenin and vinculin in the absence of α -catenin, further corroborating their findings. This data, along with the comprehensive evidence from super-resolution microscopy, FRAP, laser ablation, and co-immunoprecipitation, convincingly establishes the novel role of β -catenin in force transmission at cadherin-based adhesions.

Regarding the previous concern about vinculin expression levels in different cell lines and their potential impact on the observed z-displacement, the authors have adequately addressed this by analyzing the relative distribution of vinculin at the cell-cell junctions versus the total vinculin. By analyzing the relative distribution of vinculin at the cell-cell junctions versus the total vinculin, they have demonstrated that the lower recruitment of the vinculin T12-A50I mutant to cadherin-mediated adhesions compared to wild-type vinculin and the T12 mutant is not due to differences in expression levels.

While the lack of experimental validation of the molecular dynamics simulations for the specific residues mediating α -catenin-vinculin interactions remains a limitation, the authors have acknowledged this and emphasized the need for future experimental validation. Considering the extensive experimental evidence supporting the main conclusions of the study, I believe that this limitation does not detract from the overall impact of the work.

We wish to thank all 4 reviewers for raising important thought-provoking questions that have greatly improved the quality of the manuscripts and strengthened the scientific foundation of our work. We are also grateful for recognizing the importance of our findings.